# Revisiting Hallucination Detection with Effective Rank-based Uncertainty

## Abstract

Detecting hallucinations in large language models (LLMs) remains a fundamental challenge for their trustworthy deployment. Going beyond basic uncertainty-driven hallucination detection frameworks, we propose a simple yet powerful method that quantifies uncertainty by measuring the effective rank of hidden states derived from multiple model outputs and different layers. Grounded in the spectral analysis of representations, our approach provides interpretable insights into the model's internal reasoning process through semantic variations, while requiring no extra knowledge or additional modules, thus offering a combination of theoretical elegance and practical efficiency. Meanwhile, we theoretically demonstrate the necessity of quantifying uncertainty both internally (representations of a single response) and externally (different responses), providing a justification for using representations among different layers and responses from LLMs to detect hallucinations. Extensive experiments demonstrate that our method effectively detects hallucinations and generalizes robustly across various scenarios, contributing to a new paradigm of hallucination detection for LLM truthfulness.

## 1 Introduction

The advent of Large Language Models (LLMs) has catalyzed a paradigm shift across artificial intelligence, enabling remarkable capabilities in generative and reasoning tasks. From sophisticated dialogue to complex code generation, their prowess is undeniable (Achiam et al., 2023; Team et al., 2023). However, their reliability continues to suffer from **hallucinations** (Huang et al., 2025), where models produce outputs that are fluent and contextually plausible, but factually incorrect. Unlike simple errors, hallucinations are particularly insidious because they are often indistinguishable from trustworthy responses, making them especially dangerous in high-stakes domains such as healthcare and scientific discovery. The issue arises from the probabilistic nature of next-token prediction: LLMs optimize for linguistic plausibility rather than factual accuracy, and without explicit grounding, minor biases or gaps in knowledge can cascade into confident yet misleading narratives. As a result, hallucinations not only limit practical deployment but also raise fundamental questions about the epistemic reliability of generative models.

Despite various attempts to mitigate hallucinations through architectural modifications and training heuristics, the issue remains far from solved. The challenge of mitigating hallucinations is intrinsically tied to the problem of **uncertainty quantification (UQ)**. Ideally, an LLM should not only provide an answer but also communicate how confident it is. If a model could reliably estimate its epistemic uncertainty, it could abstain or defer when unsure, thereby reducing hallucinations (Kendall & Gal, 2017; Abbasi Yadkori et al., 2024). Although a large body of research has explored UQ in traditional machine learning, existing methods such as Monte Carlo dropout (Gal & Ghahramani, 2016) or deep ensembles (Lakshminarayanan et al., 2017) are computationally impractical for billion-parameter LLMs. Similarly, approaches based on retrieval augmentation or auxiliary calibration modules add system complexity and latency. This highlights the pressing need for lightweight, self-contained, and scalable UQ techniques tailored to the architecture and deployment constraints of modern LLMs.

In this paper, we propose an **internally interpretable** solution for UQ that operates purely from the internal state of the model. Our hypothesis is that the internal representations of LLMs contain rich but underutilized information. These representations not only reveal the model's reasoning process, enhancing internal interpretability, but their sampling probability distributions also capture

different reasoning paths, which in turn manifest as semantic divergences. Intuitively, since LLMs inherently involve stochasticity during forward propagation, small probabilistic perturbations can lead to noticeable shifts in their generation process. When a model lacks sufficient knowledge or reasoning ability, these perturbations can cause its internal representations to diverge semantically during layer-wise forward passes and eventually surface as hallucinations (Huang et al., 2025). In contrast, a confident and well-grounded model maintains robust and constrained hidden trajectories despite such perturbations, leading to consistent and reliable output. Thus, representational divergence provides a natural signal for quantifying uncertainty and identifying hallucination-prone generations.

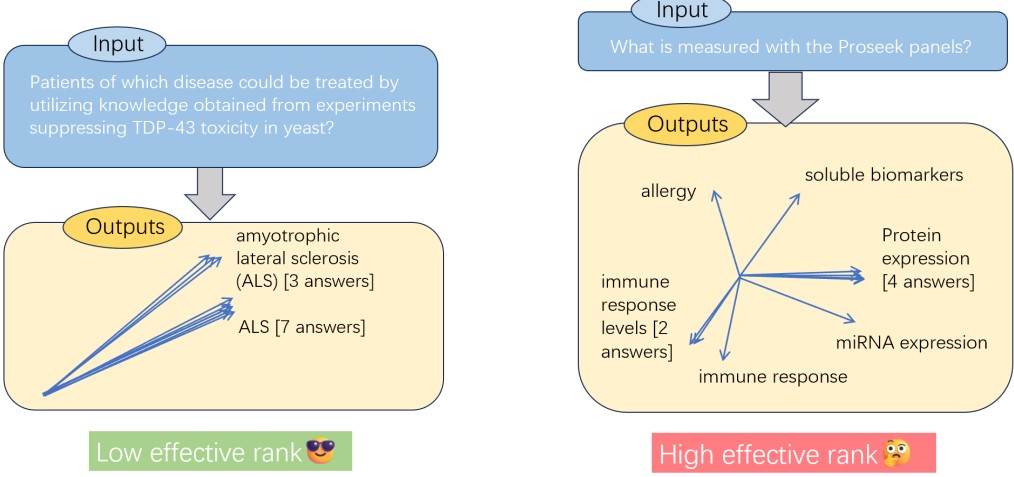

(a) The correct answers with low uncertainty      (b) The hallucinated answers with high uncertainty

Figure 1: Examples of detecting hallucinations using Effective Rank-based Uncertainty.

To formalize this intuition, we leverage the notion of **effective rank** (Roy & Vetterli, 2007) of the matrix of embedding vectors as a measure of uncertainty. Effective rank serves as a smooth measure of the divergence among vectors within a matrix, and is employed in our method to detect semantic variations in different embedding vectors. Intuitively, a low effective rank indicates that the activation energy is concentrated in a few directions, suggesting a confident, deterministic internal state (Figure 1a). By contrast, a high effective rank signals that the energy is spread diffusely, indicative of uncertainty, confusion, and a higher likelihood of hallucination (Figure 1b).

We extensively validate our hypothesis across multiple benchmark datasets and model architectures. Our experiments demonstrate that effective rank is a highly predictive signal for hallucination detection, consistently outperforming or matching the performance of established strong baselines while being significantly efficient and scalable. Additionally, we demonstrate how aleatoric uncertainty (the inherent stochasticity of LLMs) progressively amplifies and obscures epistemic uncertainty (uncertainty in the knowledge and capabilities encoded by the model's parameters) within the model's internal representations during a single sequence generation without multiple samples. This finding provides both the theoretical justification for the necessity of semantic sampling through multiple responses and the foundational rationale for its effectiveness.

Our contributions are threefold:

- **A Novel Spectral Perspective on Uncertainty**. We propose and empirically validate the Effective Rank-based Uncertainty as an efficient, robust, and model-intrinsic measure of uncertainty for white-box LLMs.

- **A lightweight, Training-Free Detector**. We introduce a rapid hallucination detection method that does not require additional tools, fine-tuning, or external knowledge.

- **Practical and Theoretical Analysis**. We conducted experiments across different scenarios, demonstrating the consistent effectiveness of our method and providing interpretable insights for the quantitative analysis of uncertainty within the model's internal representations.

## 2 RELATED WORK

### 2.1 HALLUCINATION DETECTION

Hallucination in LLMs has been widely studied from multiple perspectives. A major line of work attributes hallucinations to various forms of uncertainty inherent in the model. Specifically, hallucinations often arise due to (i) insufficient or incomplete knowledge and reasoning capabilities, (ii) excessive stochasticity in the generation process, and (iii) low faithfulness to the provided context. These uncertainty-related factors account for a large proportion of hallucination cases in practice (Huang et al., 2025; Zhang et al., 2025; Tonmoy et al., 2024). However, hallucinations can also occur even when model uncertainty is low, for example, when the model's internal knowledge itself is systematically incorrect or outdated. Such cases typically require techniques like knowledge editing or targeted fine-tuning for correction (Huang et al., 2024).

To correct and reduce hallucinations in LLMs, multiple kinds of hallucination detection methods are designed, which mainly fall into three categories. Retrieval-based methods verify generated content against external knowledge sources or retrieved evidence (Lewis et al., 2020). Self-verification approaches such as P_False (Kadavath et al., 2022) and SelfCheckGPT (Manakul et al., 2023) prompt the model to evaluate its own generations for consistency or correctness. Classifier-based detection involves training classification models on model hidden states to identify hallucinated content (Arteaga et al., 2024; Su et al., 2024; Du et al., 2024; Park et al., 2025). For example, **MIND** (Su et al., 2024) trains a small MLP on hidden states recorded when an LLM continues truncated Wikipedia passages. By learning whether the continuation mentions the correct entity, MIND predicts hallucinations directly from internal activations with low computational cost. While often effective, these methods typically depend on extra knowledge and additional training, and may introduce non-negligible latency during inference. These limitations motivate the exploration of intrinsic uncertainty-based detection paradigms.

### 2.2 UNCERTAINTY QUANTIFICATION FOR HALLUCINATION DETECTION

Uncertainty estimation has emerged as a prominent approach for hallucination detection in LLMs Zhang et al. (2023). The core premise is that a model's internal confidence can indicate output veracity. Initial approaches used token-level or sequence-level probability distributions, such as the **Length-normalized Entropy** (Malinin & Gales, 2020), which normalizes entropy by the number of tokens to mitigate the impact of varying sequence lengths. However, these methods capture lexical rather than semantic uncertainty. As models can be lexically confident yet semantically incorrect (Kuhn et al., 2023), recent benchmarks (Ye et al., 2024; Nado et al., 2021) have spurred exploration of semantic properties and internal representations for improved detection.

Based on semantic analysis, Kuhn et al. (2023); Farquhar et al. (2024) introduced **Semantic Entropy** (SE) and **Discrete Semantic Entropy** (DSE) to address limitations of token-level metrics. SE and DSE cluster generations by semantic equivalence using natural language inference (NLI) models, then compute entropy over the distribution of semantic categories. Note that while DSE directly estimates the probability of each semantic category using the frequency of answers in each cluster, SE further incorporates a token-level probability adjustment. High SE indicates semantic uncertainty, effectively capturing meaning-level variations. Meanwhile, the INSIDE (Chen et al., 2024) measures the model's internal uncertainty by extracting the internal embedding vectors of the model, computing their covariance matrix, and calculating the determinant of the covariance matrix through eigenvalue decomposition to obtain the **Eigenscore**. This Eigenscore approximates *differential entropy*, with higher scores indicating internal inconsistency. These methods each delve deeply into the uncertainty about semantic variances or internal representations, but the relationship between internal representations and external semantic information remains largely unexplored.

## 3 METHODOLOGY

In this section, we propose a novel uncertainty-based approach to hallucination detection in LLMs. Our primary motivation is to leverage the model's internal representations to measure its uncertainty, and we adopt the effective rank from spectral analysis as our main method, which enjoys smoothness and clear practical interpretability.

## 3.1 CONSTRUCTING THE EMBEDDING MATRIX

For each query $q$, we first construct the representation matrix extracted from different layers and responses to calculate the effective rank. To this end, we sample $m_1$ responses and extract embeddings from $m_2$ layers per response, resulting in $m = m_1 \times m_2$ embedding vectors $\vec{a_1}, \vec{a_2}, \cdots, \vec{a_m} \in \mathbb{R}^n$. These vectors are concatenated into a representation matrix:

$$A = [\vec{a_1}, \vec{a_2}, \cdots, \vec{a_m}] \in \mathbb{R}^{n \times m} \tag{1}$$

where each $\vec{a_i}$ corresponds to the embedding of a particular response at a specific layer. Following Skean et al. (2025), who find that intermediate layers of an LLM strike the best balance between preserving useful information and removing noise, while the final layer often overfits the training objective and eliminates generalization, we therefore extract the embedding vector from the exact middle hidden layer for every response, both in our method and the Eigenscore baseline.

## 3.2 SPECTRAL ANALYSIS

Next, we want to quantify the dispersion of the column vectors of this matrix, and the notion of singular values offers a perfectly suited mathematical tool for this, since singular values measure how much the matrix stretches or compresses vectors along certain directions in space. We then compute the singular value decomposition (SVD) of the matrix: $A = U\Sigma V^\top$, where $\Sigma = \text{diag}(\sigma_1, \sigma_2, \ldots, \sigma_m)$ contains the singular values in descending order.

To quantify the dispersion of vectors across different directions using the entropy of the singular values, we normalize the singular value spectrum into a probability distribution:

$$p_i = \frac{\sigma_i}{\sum_{j=1}^{m} \sigma_j}, \quad i = 1, \ldots, m \tag{2}$$

## 3.3 ENTROPY-BASED MEASURE AND EFFECTIVE RANK

We consider the Shannon entropy of the singular value distribution as our measure of uncertainty for the query $q$:

$$H = -\sum_{i=1}^{m} p_i \ln p_i \tag{3}$$

Using entropy to quantify uncertainty is common, yet our approach rests on an exceptionally clear and elegant mathematical foundation. The quantity $\exp(H)$ is the *effective rank* of the matrix $A$, which measures the "effective linear independence" of the vectors in matrix $A$ (Roy & Vetterli, 2007). Intuitively, $\exp(H)$ corresponds to the effective number of distinct semantic modes represented in the embedding vectors. Hence, a larger entropy indicates higher uncertainty in the model's outputs.

Through Jensen's inequality, it can be readily shown that the effective rank of a matrix equals its true rank if and only if the matrix has a rank of 1 or its column vectors are pairwise orthogonal and of equal length (corresponding to the cases of minimum and maximum uncertainty, respectively). In all other cases, the effective rank is strictly less than the true rank and varies continuously with the dispersion of the vectors. Therefore, effective rank smoothly encodes how divergent the column vectors are. In practice, semantically similar embedding vectors are often similar yet not perfectly aligned, so a continuous effective rank captures their consistency far better than the discrete true rank. Moreover, the effective rank here possesses a clear and practical interpretation as the "effective number of semantic categories", providing us with an excellent tool for further analyzing and enhancing the internal interpretability of LLM models (Zhuo et al., 2023).

We summarize the above procedure for computing the effective rank in Algorithm 1. In the following section, we will empirically validate the effectiveness, robustness and generality of this method.

## 4 EXPERIMENTS

### 4.1 EXPERIMENTAL SETUP

**Datasets.** Our dataset selection is designed to rigorously evaluate hallucination detection across a spectrum of knowledge and reasoning domains. TriviaQA (Joshi et al., 2017) serves as a broad

---

**Algorithm 1:** Effective Rank-based Uncertainty Computation

---

**Input:** Query $q$, number of responses $m_1$, number of layers per response $m_2$, LLM (white-box)
**Output:** Effective rank $\exp(H)$
**for** $i : 1 \to m_1$ *(in parallel)* **do**
    Sample response $r_i$ from the LLM using query $q$;
    Extract the hidden state $h_i$ corresponding to the last generated token of response $r_i$;
    **for** $j : 1 \to m_2$ *(in parallel)* **do**
        Extract embedding vector $\mathbf{a}_{ij} \in \mathbb{R}^n$ from the corresponding position of $h_i$
    **end**
**end**
Form embedding matrix $A = [\mathbf{a}_{11}, \ldots, \mathbf{a}_{1m_2}, \ldots, \mathbf{a}_{m_1 m_2}] \in \mathbb{R}^{n \times m}$, where $m = m_1 \times m_2$;
Compute SVD: $A = U \Sigma V^\top$, with singular values $\{\sigma_1, \ldots, \sigma_m\}$;
Normalize singular values: $p_i = \sigma_i / \sum_{j=1}^m \sigma_j$ for $i = 1, \ldots, m$;
Compute entropy: $H = -\sum_{i=1}^m p_i \ln p_i$;
**return** $\exp(H)$

---

open-domain knowledge benchmark. In contrast, Natural Questions (NQ) (Kwiatkowski et al., 2019) and BioASQ (Tsatsaronis et al., 2015) probe deeper into specialized knowledge within natural science and healthcare domains where LLM hallucinations can be particularly misleading and harmful. To test foundational competencies beyond factual recall, we also include SQuAD (Rajpurkar et al., 2016) for context understanding and text inference.

**Models.** For our backbone language models, we utilize three widely recognized open-weight models: Llama-2-7b-chat (Touvron et al., 2023), Llama-2-13b-chat, and Mistral-7B-v0.1 (Jiang et al., 2023) to evaluate our methods across models of varying scales and distinct architectural lineages.

**Evaluation Metrics.** To determine whether a model's generation contains hallucinations, we use **ROUGE-L** (Lin, 2004), an n-gram based metric that computes the longest common subsequence between the generation and the ground-truth answer. A generation is considered correct if its ROUGE-L score $\geq 0.5$; otherwise, it is flagged as a hallucination. The validity of this annotation method is discussed in Appendix F. To evaluate the overall performance of the hallucination detection methods themselves, we use the **Area Under the Receiver Operating Characteristic curve (AUROC)** (Filos et al., 2019). AUROC is chosen as it provides a comprehensive measure that is agnostic to the absolute scale of the uncertainty scores produced by different detectors. This makes it an ideal metric for comparing the ability to rank hallucinated examples higher than non-hallucinated ones.

**Baselines.** Beyong strong baselines including (Discrete) Semantic Entropy (**SE/DSE**) (Farquhar et al., 2024), and **Eigenscore** (Chen et al., 2024) discussed in Section 2, we additioanlly consider recent state-of-the-art methods inlcuding (Weighted) Semantic Nearest Neighbor Entropy (**SNNE/WSNNE**) (Nguyen et al., 2025), which compute the balanced pairwise similarity between different answers, and Shifting Attention to Relevance (**SAR**) (Duan et al., 2024), which incorporates attention weights based on the Predictive Entropy of the answers. We also considered these representative classical baselines: **P_False** (Kadavath et al., 2022), and **Length-Normalized Entropy** (Malinin & Gales, 2020). Details of these baselines can be found in Section 2 and Appendix D.

**Implementation Details.** All experiments were conducted on a virtual GPU (vGPU) with 48GB of memory. We use a temperature of 1.0 for sampling and generate $N = 10$ answers per input. For embedding extraction, we utilize the hidden state of the last token from the exact middle layer of the model. Finally, in the Eigenscore calculation, we follow the recommendation of Chen et al. (2024) and set the small regularization term to $\alpha = 0.001$.

## 4.2 MAIN RESULTS

The overall performance comparison of our Effective Rank (ER) method against five baselines is summarized in Table 1. The results, measured in AUROC across three models and five diverse datasets, demonstrate the efficacy and generality of our approach in detecting hallucinations.

Table 1: Overall comparison of ER with baselines, where ER denotes Effective Rank, ES denotes Eigenscore, PF denotes P_False, DSE denotes Discrete Semantic Entropy, LNE denotes Length-Normalized Entropy, and SE denotes Semantic Entropy. All numbers in the table are AUROC scores; values closer to 1 indicate stronger hallucination-detection ability. The best results are shown in **bold**, and the second-best results are underlined.

| Model | Dataset | Accuracy | ER (Ours) | ES | PF | DSE | LNE | SE |
|---|---|---|---|---|---|---|---|---|
| Llama-2-7b | TriviaQA | 0.602 | $\mathbf{0.7873}_{\pm 0.0022}$ | $\underline{0.7792}_{\pm 0.0019}$ | $0.6650_{\pm 0.0025}$ | $0.7754_{\pm 0.0015}$ | $0.6935_{\pm 0.0015}$ | $0.7746_{\pm 0.0012}$ |
| | SQuAD | 0.427 | $\mathbf{0.7211}_{\pm 0.0018}$ | $\underline{0.7197}_{\pm 0.0021}$ | $0.6613_{\pm 0.0021}$ | $0.7176_{\pm 0.0016}$ | $0.6519_{\pm 0.0008}$ | $0.7179_{\pm 0.0016}$ |
| | BioASQ | 0.305 | $\mathbf{0.8453}_{\pm 0.0016}$ | $\underline{0.8438}_{\pm 0.0011}$ | $0.7322_{\pm 0.0023}$ | $0.8433_{\pm 0.0019}$ | $0.4706_{\pm 0.0031}$ | $0.8425_{\pm 0.0012}$ |
| | NQ | 0.266 | $\underline{0.7041}_{\pm 0.0011}$ | $\mathbf{0.7064}_{\pm 0.0013}$ | $0.6589_{\pm 0.0022}$ | $0.6966_{\pm 0.0013}$ | $0.6500_{\pm 0.0012}$ | $0.6996_{\pm 0.0026}$ |
| | **Average** | 0.400 | **0.7645** | 0.7623 | 0.6794 | 0.7582 | 0.6165 | 0.7587 |
| Llama-2-13b | TriviaQA | 0.656 | $\mathbf{0.7412}_{\pm 0.0025}$ | $0.7344_{\pm 0.0019}$ | $0.7356_{\pm 0.0033}$ | $0.7328_{\pm 0.0021}$ | $0.6927_{\pm 0.0020}$ | $\underline{0.7371}_{\pm 0.0028}$ |
| | SQuAD | 0.422 | $0.7271_{\pm 0.0024}$ | $0.7246_{\pm 0.0027}$ | $0.6935_{\pm 0.0027}$ | $\mathbf{0.7366}_{\pm 0.0020}$ | $0.6894_{\pm 0.0020}$ | $\underline{0.7364}_{\pm 0.0014}$ |
| | BioASQ | 0.359 | $\mathbf{0.8234}_{\pm 0.0020}$ | $\underline{0.8213}_{\pm 0.0021}$ | $0.6951_{\pm 0.0023}$ | $0.7943_{\pm 0.0012}$ | $0.6894_{\pm 0.0018}$ | $0.7992_{\pm 0.0016}$ |
| | NQ | 0.313 | $\mathbf{0.7283}_{\pm 0.0015}$ | $\underline{0.7268}_{\pm 0.0014}$ | $0.6831_{\pm 0.0027}$ | $0.7246_{\pm 0.0016}$ | $0.6872_{\pm 0.0021}$ | $0.7248_{\pm 0.0026}$ |
| | **Average** | 0.438 | **0.7550** | 0.7517 | 0.7018 | 0.7471 | 0.6897 | 0.7494 |
| Mistral-7b | TriviaQA | 0.359 | $\mathbf{0.7635}_{\pm 0.0017}$ | $0.7575_{\pm 0.0018}$ | $0.7200_{\pm 0.0026}$ | $0.7579_{\pm 0.0012}$ | $0.6521_{\pm 0.0025}$ | $0.7559_{\pm 0.0023}$ |
| | SQuAD | 0.229 | $0.7214_{\pm 0.0021}$ | $0.7133_{\pm 0.0017}$ | $0.6381_{\pm 0.0038}$ | $\underline{0.7222}_{\pm 0.0019}$ | $0.6223_{\pm 0.0017}$ | $\mathbf{0.7236}_{\pm 0.0021}$ |
| | BioASQ | 0.018 | $\mathbf{0.8562}_{\pm 0.0024}$ | $\underline{0.8525}_{\pm 0.0015}$ | $0.7178_{\pm 0.0024}$ | $0.8505_{\pm 0.0013}$ | $0.5946_{\pm 0.0017}$ | $0.8508_{\pm 0.0026}$ |
| | NQ | 0.114 | $\underline{0.7662}_{\pm 0.0025}$ | $0.7624_{\pm 0.0015}$ | $0.7531_{\pm 0.0027}$ | $\mathbf{0.7675}_{\pm 0.0020}$ | $0.6779_{\pm 0.0014}$ | $0.7658_{\pm 0.0020}$ |
| | **Average** | 0.180 | **0.7768** | 0.7714 | 0.7073 | $\underline{0.7745}$ | 0.6367 | 0.7740 |

**Overall Effectiveness.** Our method achieves the highest AUROC score in 8 out of 12 evaluation scenarios, establishing a consistent and strong performance baseline. Notably, this superiority is not confined to a specific model or dataset but is observed across various scales (7b and 13b parameters) and architectures (Llama and Mistral). This indicates that the principle of analyzing representation space stability through effective rank is a generalizable strategy for hallucination detection. Even in cases where ER does not rank first, its performance remains highly competitive (e.g., on NQ dataset). We also note that our method's overall margin over the baselines is larger on Llama-2-13B-chat than on Llama-2-7B-chat, suggesting its potential scales with model size.

**Analysis of Shortcomings.** A primary limitation of our method is observed in its unstable performance on the SQuAD dataset. This suggests that for tasks requiring complex textual reasoning and comprehension, rather than pure factual recall, the relationship between internal representations and uncertainty may be less consistent. In such scenarios, SE and DSE methods, which directly measure semantic variation across multiple sampled answers, appear to capture uncertainty more reliably. Nevertheless, Effective Rank remains highly competitive compared to all methods except SE and DSE on SQuAD dataset. Our ablation study further analyzes the interplay between the roles of different hidden layers and the requirements of tasks across various datasets.

**Comparative Analysis of Baselines.** The results also shed light on the relative performance of existing methods. The Eigenscore method, apart from showing a noticeable gap compared to Effective Rank on the TriviaQA dataset, remains relatively close to our approach and occasionally even surpasses it by a narrow margin. This is because both methods utilize the representations from the model's internal hidden layers to measure uncertainty. However, Eigenscore is an approximation of differential entropy, whereas Effective Rank directly yields an intuitive and effective uncertainty indicator, the "effective number of distinct semantic categories". This gives Effective Rank an overall advantage over Eigenscore. Although SE and DSE methods generally perform better than our approach on SQuAD, they require an auxiliary NLI model and additional time for semantic reasoning (Table 2). Additionally, while P_False and LNE methods are relatively unstable, they occasionally deliver relatively strong performance.

**Evaluation on advanced tasks.** Beyond these benchmarks assessing the factual question answering and textual reasoning of LLMs, we additionally consider more advanced datasets like long-form generation (CoQA (Reddy et al., 2019) on multi-turn QA, CNN DailyMail (Hermann et al., 2015) on summarization), complex reasoning (e.g., MATH-500 (Lightman et al., 2023) on multi-step math problems), and agentic workflows (e.g., Humaneval (Chen, 2021) on coding). In our experiments, we use the cosine similarity of the final hidden-layer embedding vectors to compute the pairwise similarity for SNNE/WSNNE, and we use a Cross-Encoder–based approach to measure the influence of removing each token from the answer as the attention weight (here we only use token-level attention, rather than combining both token- and sentence-level attention as in the full SAR method).

Table 2: Qualitative comparison of hallucination detection methods. In terms of time efficiency, the numbers in the table indicate the average time spent on each question in the experiment where each method generated results for 3 models × 4 datasets under the parameter settings of the main experiment. Our method took an average of 9.5 seconds, which is essentially the same as the time required to naturally generate answers without hallucination detection. The detailed time complexity analysis of our ER method can be found in Appendix F.

| Methods | Effective | Internally interpretable | Semantic Information | Time Efficiency |
|---------|-----------|--------------------------|----------------------|-----------------|
| ES | ✓ | ✓ | ✗ | 9.5s |
| SE, DSE | ✓ | ✗ | ✓ | 11.7s |
| PF | ✗ | ✗ | ✗ | 10.1s |
| LNE | ✗ | ✗ | ✗ | 9.6s |
| ER (ours) | ✓ | ✓ | ✓ | 9.5s |

From Table 3, we observe that our method maintains a significant lead on the coding and summarization tasks, while it is slightly weaker than SE/DSE on mathematical reasoning (where SE/DSE denotes the larger AUROC value obtained using either the SE or DSE method). This disadvantage is likely due to the semantic mismatch between the LLM's hidden-layer embeddings and the final answer in mathematical reasoning, whereas SE and DSE compute semantic uncertainty directly from the generated answers. In addition, for these advanced tasks, the responses are much longer and more complex than those in earlier datasets, making ROUGE-L an unreliable metric for evaluating correctness. Therefore, on these datasets, we rely on prompts that ask the model to judge correctness based on both its answer and the standard answer.

Table 3: Evaluation on advanced reasoning tasks. The best results are shown in **bold**, and the second-best results are underlined.

| Model | Task | Dataset | ER (Ours) | ES | PF | SE/DSE | LNE | SNNE | WSNNE | SAR |
|-------|------|---------|-----------|-----|-----|--------|-----|------|-------|-----|
| Llama-2-7b | Multi-turn QA | CoQA | **0.7339** | 0.7316 | 0.6471 | 0.7284 | 0.6979 | 0.7208 | 0.7235 | 0.7148 |
| | Math | MATH-500 | 0.6958 | 0.6910 | 0.5518 | **0.7026** | 0.6111 | 0.6775 | 0.6714 | 0.6924 |
| | Coding | HumanEval | **0.6268** | 0.6209 | 0.4931 | 0.5564 | 0.5152 | 0.5983 | 0.5965 | 0.6010 |
| | Summarization | CNN Dailymail | **0.6880** | 0.6794 | 0.4276 | 0.6539 | 0.5596 | 0.6697 | 0.6641 | 0.6324 |
| | | **Average** | **0.6861** | 0.6807 | 0.5299 | 0.6603 | 0.5960 | 0.6666 | 0.6639 | 0.6602 |
| Llama-3-8b | Multi-turn QA | CoQA | **0.7529** | 0.7441 | 0.5907 | 0.7433 | 0.6159 | 0.7472 | 0.7448 | 0.7276 |
| | Math | MATH-500 | 0.7071 | 0.6994 | 0.6247 | **0.7104** | 0.6031 | 0.6951 | 0.6975 | 0.6974 |
| | Coding | HumanEval | **0.6217** | 0.6173 | 0.5072 | 0.5822 | 0.4918 | 0.6117 | 0.6158 | 0.6103 |
| | Summarization | CNN Dailymail | **0.6359** | 0.6200 | 0.4373 | 0.6040 | 0.5482 | 0.6133 | 0.6089 | 0.5904 |
| | | **Average** | **0.6794** | 0.6702 | 0.5400 | 0.6600 | 0.5648 | 0.6668 | 0.6668 | 0.6564 |
| Qwen3-8B | Multi-turn QA | CoQA | 0.7788 | **0.7797** | 0.6482 | 0.7715 | 0.6540 | 0.7613 | 0.7642 | 0.7509 |
| | Math | MATH-500 | 0.7319 | 0.7298 | 0.4770 | **0.7375** | 0.6466 | 0.6900 | 0.6981 | 0.7225 |
| | Coding | HumanEval | **0.6479** | 0.6352 | 0.4736 | 0.5734 | 0.5834 | 0.6387 | 0.6331 | 0.6381 |
| | Summarization | CNN Dailymail | **0.6880** | 0.6794 | 0.4276 | 0.6539 | 0.5596 | 0.6697 | 0.6641 | 0.6453 |
| | | **Average** | **0.7117** | 0.7060 | 0.5066 | 0.6841 | 0.6109 | 0.6899 | 0.6899 | 0.6892 |

In conclusion, the main results strongly support the effectiveness of our Effective Rank approach. Its dominant performance on TriviaQA and BioASQ tasks highlights its value as a tool for detecting factual hallucinations, a critical challenge for modern LLMs.

### 4.3 ABLATION STUDIES

**Number of generations and the use of different hidden layers.** Our ablation studies on Llama-2-7b-chat reveal that the performance of the ER method is robust to the number of generations, since it maintains a consistent overall advantage over the baselines across different values of N. Moreover, no single layer extraction strategy consistently outperforms all others, indicating a complex interaction between the task domain and the representational properties of different model layers. Notably, although M5, L1, and L5 hidden layer selection strategies yield results comparable to M1 and can complement M1 to a certain extent, they are slightly inferior overall and exhibit marginally less stability (somewhat analogous to how DSE is generally slightly weaker than SE). Furthermore, increasing the number of generations (N) does not yield uniform improvements, implying a task-dependent optimum exists for this hyperparameter.

Table 4: Ablation studies on Llama-2-7b-chat, where N refers to the number of generations, M1 refers to extracting the exact middle hidden layer for each answer in the ER method (i.e., the ER method used in the main experiment), M5 refers to extracting the middle five layers, L1 refers to extracting the last layer, and L5 refers to extracting the last five layers. Ablation studies on other models and complete experimental results can be found in Appendix E.

| N | Dataset | M1 | M5 | L1 | L5 | Best Baseline |
|---|---------|-----|-----|-----|-----|---------------|
| 10 | TriviaQA | **0.7877** | 0.7700 | 0.7809 | 0.7629 | 0.7802 (ES) |
| | SQuAD | 0.7212 | **0.7217** | 0.7191 | 0.7190 | 0.7199 (ES) |
| | BioASQ | 0.8447 | 0.8461 | **0.8481** | 0.8472 | 0.8437 (DSE) |
| | NQ | 0.7029 | 0.7038 | 0.7036 | 0.7049 | **0.7056** (ES) |
| | Average | **0.7641** | 0.7604 | 0.7629 | 0.7585 | 0.7623 (ES) |
| 15 | TriviaQA | 0.7862 | **0.7902** | 0.7870 | 0.7807 | 0.7825 (ES) |
| | SQuAD | **0.7407** | **0.7407** | 0.7391 | 0.7395 | 0.7391 (ES) |
| | BioASQ | **0.8715** | 0.8690 | 0.8684 | 0.8678 | 0.8682 (ES) |
| | NQ | 0.7182 | 0.7166 | 0.7174 | 0.7188 | **0.7206** (ES) |
| | Average | **0.7792** | 0.7780 | 0.7767 | 0.7791 | 0.7776 (ES) |
| 20 | TriviaQA | **0.7786** | 0.7780 | 0.7729 | 0.7702 | 0.7743 (ES) |
| | SQuAD | 0.7596 | 0.7604 | 0.7599 | **0.7610** | 0.7588 (DSE) |
| | BioASQ | **0.8645** | 0.8625 | 0.8638 | 0.8620 | 0.8628 (ES) |
| | NQ | 0.7277 | **0.7292** | 0.7274 | 0.7278 | 0.7284 (ES) |
| | Average | **0.7826** | 0.7825 | 0.7810 | 0.7803 | 0.7809 (ES) |

Table 5: Ablation studies on temperature (denoted as $t$), and Best ER refers to the Effective Rank that performed the best among the four hidden vector selection methods (M1, M5, L1, L5). Complete experiment results can be found in Appendix E.

| | | TriviaQA | SQuAD | BioASQ | NQ | Average |
|---|---|----------|-------|--------|-----|---------|
| $t = 0.1$ | Best ER | **0.6782** (L1) | **0.6466** (M5) | 0.7416 (L5) | 0.6841 (L1) | 0.6768 (L1) |
| | Best Baseline | 0.6695 (PF) | 0.6413 (PF) | **0.7720** (PF) | **0.7308** (PF) | **0.7034** (PF) |
| $t = 0.5$ | Best ER | **0.7628** (M5) | **0.7401** (L5) | **0.8452** (L5) | **0.7724** (L1) | **0.7771** (L1) |
| | Best Baseline | 0.7350 (ES) | 0.7330 (ES) | 0.8378 (ES) | 0.7641 (ES) | 0.7675 (ES) |
| $t = 1.0$ | Best ER | **0.7651** (L1) | 0.7208 (M1) | **0.8584** (M5) | **0.7696** (M5) | **0.7769** (M5) |
| | Best Baseline | 0.7579 (ES) | **0.7268** (SE) | 0.8513 (ES) | 0.7678 (DSE) | 0.7747 (DSE) |
| $t = 2.0$ | Best ER | 0.6685 (M5) | **0.5754** (M1) | **0.7329** (M5) | 0.6118 (M1) | 0.6437 (M1) |
| | Best Baseline | **0.7084** (PF) | 0.5698 (ES) | 0.7244 (SE) | **0.7006** (PF) | **0.6640** (PF) |

**Temperature.** To assess the impact of decoding temperature $t$ on detection performance, we evaluate the detection AUROC under Mistral-7B and report the results in Table 5, where we observe that the performance is generally optimal when the temperature is set around 0.5 and 1.0. Under these conditions, our Effective Rank method also demonstrates a significant overall advantage. On the other hand, excessively high or low temperatures substantially undermine uncertainty-based methods, while the Self-Verification-based P_False method unexpectedly exhibits a notable relative advantage in such scenarios, albeit with remaining instability. Moreover, the detailed data show that the L1 and L5 strategies are more robust to low temperatures, while the M1 and M5 strategies are more robust to high temperatures.

## 5 INTERNAL INTERPRETABILITY: THE DECOMPOSITION AND QUANTITATIVE ANALYSIS OF UNCERTAINTY

### 5.1 EMPIRICAL MOTIVATION

While multi-sample methods are effective for hallucination detection, we still want to explore the real-time detection without multiple samples. On Llama-2-7b-chat, we use a sliding window to measure the effective rank of the internal representations across every three consecutive layers and then take

the average. The final results vary slightly between different responses, but generally fluctuate around 1.9. This indicates that semantic changes occur within the model during the reasoning process, and these shifts exhibit a certain degree of similarity but little variability between different responses. This finding is broadly consistent with conclusions drawn from using cosine similarity of internal representations to assess internal consistency (Min et al., 2024). However, the experiments revealed that this form of internal uncertainty showed only a weak correlation with hallucinations (overall AUROC $\approx 0.57$, barely exceeding the random chance). This unsuccessful attempt prompted us to reconsider: *What truly constitutes effective internal uncertainty for hallucination detection?*

According to the perspective of Depeweg et al. (2018), the uncertainty in the predictions of Deep Neural Networks can be decomposed into two primary types: *aleatoric uncertainty*, which is inherent in the data and model architecture due to its ambiguity or inherent stochasticity, and *epistemic uncertainty*, which stems from the model's lack of knowledge or ability. The latter is considered particularly critical for detecting hallucinations. We subsequently argue, however, that within any single sampled response, the information signal pertaining to epistemic uncertainty is largely masked by aleatoric uncertainty. Furthermore, this aleatoric uncertainty can propagate and become amplified through the model's reasoning process. Consequently, when the LLM's internal knowledge is insufficient to produce a determinate result amidst this prevailing uncertainty noise (i.e., when a hallucination occurs), the generation and reasoning processes of LLMs often involve probabilistic distributions with obvious uncertainty, which ultimately manifest as semantic divergences across different sampled answers (Manakul et al., 2023; Huang et al., 2025).

## 5.2 PRELIMINARIES AND PROBLEM SETUP

Consider an autoregressive language model parameterized by $\theta$. Given an input prompt $q$, the model generates a sequence of tokens $(y_1, y_2, \ldots, y_T)$ and a corresponding sequence of hidden states $(h_1, h_2, \ldots, h_L)$, where $h_t \in \mathbb{R}^d$ is the hidden state at step $t$. The generation process is defined as:

$$y_t \sim p(y_t|h_{t-1};\theta), \quad h_t = f(h_{t-1}, y_t; \theta)$$

where $f$ is a deterministic, non-linear transformation, and $p$ is the model's output distribution.

To frame our discussion, we adapt the Bayesian deep learning framework (Kendall & Gal, 2017; Depeweg et al., 2018) to reason about the hidden states. We consider the expected total variance of the hidden state $h_t$ across the data generation process, which can be decomposed into two components:

$$\underbrace{\text{Var}(h_t)}_{\text{Total Uncertainty}} = \underbrace{\mathbb{E}_\theta[\text{Var}(h_t|\theta)]}_{\text{Aleatoric Uncertainty}} + \underbrace{\text{Var}_\theta(\mathbb{E}[h_t|\theta])}_{\text{Epistemic Uncertainty}}. \tag{4}$$

Since in modern large language models, parameters are typically fixed at a point estimate, this variance decomposition relies on a Bayesian treatment of model parameters, such as deep ensembles, Monte Carlo dropout, or variational Inference. We build upon two common observations from prior work, noting that they are tendencies rather than absolute laws:

**Peaked Parameter Posterior:** In many well-trained LLMs, the parameters are often found in a region of a peaked posterior distribution $p(\theta|\mathcal{D})$ (Izmailov et al., 2018; Fort et al., 2019). This suggests that the parameter covariance $\Sigma_\theta = \text{Cov}(\theta)$ is often small, though not negligible, especially for under-regularized or smaller models.

**Representational Expansion:** Each transformer layer actively transforms and enriches its input, creating overcomplete representations in high dimensions (Sanford et al., 2023). This process inherently amplifies minor input variations, causing stochasticity to accumulate across layers (Schoenholz et al., 2016). However, this expansive potential is also constrained by key stabilizing mechanisms within the Transformer architecture, such as residual connections and layer normalization.

## 5.3 POTENTIAL DOMINANCE OF ALEATORIC UNCERTAINTY

The autoregressive process can be viewed as a Markov chain where sampling noise may accumulate through the network's dynamics. We analyze the variance at step $t$ for a fixed $\theta$ to gain intuition.

**Lemma 1** (Variance Propagation with Expansion Property). *For a fixed parameter $\theta$, the variance of the hidden state $h_t$ can be bounded from below recursively. Assuming the transformation $f$ exhibits*

*representational expansion in deep Transformers (Wang et al., 2022; Schoenholz et al., 2016), the conditional variance satisfies:*

$$\mathbb{E}_\theta[Var(h_t|\theta)] \geq \mathbb{E}_\theta \mathbb{E}_{h_{t-1}} \left[ |J_y(h_{t-1})|_F^2 \cdot Var(y_t|h_{t-1}; \theta) \right] + \Delta_{nonlin} \tag{5}$$

*where $J_y = \frac{\partial f}{\partial y}$ is the Jacobian of the transformation with regard to the input token embedding, $|\cdot|_F$ denotes the Frobenius norm, and $\Delta_{nonlin}$ is a small term capturing higher-order effects.*

**Lemma 2** (Epistemic Uncertainty Bound)**.** *The epistemic variance can be bounded by:*

$$Var_\theta(\mathbb{E}[h_t|\theta]) \leq \|G_t\|_F^2 \cdot Tr(\Sigma_\theta) + \epsilon_t \tag{6}$$

where $G_t = \frac{\partial \mathbb{E}[h_t|\theta]}{\partial \theta}$ is the sensitivity matrix of the expected hidden state trajectory to parameters, $\Sigma_\theta$ is the covariance of the parameter posterior, and $\epsilon_t$ is a non-linearity term in parameter space.

For detailed derivation and further discussion, please refer to Appendix C. Based on these lemmas, it is straightforward to obtain the final conclusion, noting its conditional nature.

**Proposition 1.** *For an autoregressive language model exhibiting a sufficiently peaked parameter posterior and representational expansion over $t$ steps, the aleatoric uncertainty in its hidden representations may dominate the epistemic uncertainty:*

$$\mathbb{E}_\theta[Var(h_t|\theta)] \gg Var_\theta(\mathbb{E}[h_t|\theta]). \tag{7}$$

### 5.4 BRIDGING THEORY AND METHOD: THE NECESSITY OF A UNIFIED APPROACH

Our theoretical analysis underscores a critical challenge: the predominance of aleatoric uncertainty during a single forward pass without multiple samples obscures the epistemic uncertainty crucial for hallucination detection. With multiple samples, we can more easily probe the different paths the model considers, approaching the full picture of its internal probability distribution. This lets us externalize that distribution as semantic divergence, yielding richer and more visible uncertainty information for hallucination detection. However, this raises the question of how to best quantify this uncertainty. Relying solely on external metrics, such as the semantic entropy of final outputs, discards the rich information within the model's internal representations. Conversely, methods that focus purely on internal statistics, such as the Eigenscore which approximates the differential entropy of internal probability distribution, lack clear and interpretable associations with external semantic information. Meanwhile, our proposed method provides exactly such a unified view. By applying effective rank to hidden state embeddings collected from multiple sampled responses, it simultaneously (i) captures the "effective number of semantic categories" emerging externally across generations and (ii) encodes the dispersion of internal representations that reflect the probabilistic dynamics of the model.

In this way, Effective Rank-based Uncertainty bridges the gap between external semantic variation and internal interpretability. The effective rank method elegantly captures how the model's internal representations continuously and interpretably transform into external semantic divergences. Therefore, we believe that further employing the effective rank method to study the external real-world meanings of internal representations is a highly promising direction.

## 6 CONCLUSION

In this work, we introduce the concept of Effective Rank-based Uncertainty, a spectral analysis tool, and apply it to the internal representations of LLMs, thereby proposing a simple and efficient method for hallucination detection. Through comprehensive experiments and ablation studies, we demonstrated the effectiveness of our approach. The practical interpretation of effective rank as the "effective number of distinct semantic categories in hidden representations" provides a clear and motivated basis for its application. Furthermore, by decomposing and quantitatively analyzing internal uncertainty, we provide in-depth theoretical modeling and analysis of the propagation of internal uncertainty noise in LLMs and how to leverage such uncertainty for hallucination detection. We believe that further exploration of effective rank as an elegant and powerful theoretical tool holds significant promise for enhancing the interpretability and reliability of AI systems.

## ETHICS STATEMENT

Our overarching goal is to contribute to the development of safer AI systems by providing users with tools to better interpret the confidence and reliability of the output of the language model. In principle, such mechanisms could mitigate several risks associated with LLMs based on foundation models, including the production of misleading or harmful content in sensitive domains such as medicine. Nevertheless, this promise also carries potential downsides: systematic errors in uncertainty estimation, or poor communication of uncertainty, could foster misplaced trust and lead to unintended harms. Although our work advances methodological understanding and introduces new approaches for uncertainty quantification in LLM, we emphasize that deployment should be preceded by a careful, context-specific evaluation to ensure that the communicated uncertainty genuinely empowers users rather than creating confusion or false assurance.

## REPRODUCIBILITY STATEMENT

Due to the high computational demands of experimentation with large foundation models, most prior work on uncertainty in LLM has depended on proprietary systems, costly fine-tuning procedures, or labor-intensive human evaluation, which can limit accessibility for many academic researchers. In contrast, our approach leverages recently released, openly available models and constructs a fully open-source pipeline for uncertainty quantification in LLM. Importantly, our method requires neither fine-tuning nor additional training of foundation models, and can be applied directly to pre-trained models in an "off-the-shelf" manner. We hope this design lowers barriers for future research in the academic community and facilitates straightforward replication of our experiments.

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

## A    THE USE OF LARGE LANGUAGE MODELS (LLMs)

In preparing this manuscript, we made limited use of LLMs solely to aid in polishing the writing. Specifically, LLMs were used for improving grammar, clarity, and style of exposition. All conceptual development, technical contributions, theoretical results, and experimental analyses were conceived and carried out entirely by the authors. The scientific content, data analysis, and conclusions remain exclusively the responsibility of the authors.

## B    MORE THEORETICAL BACKGROUND ON EFFECTIVE RANK

A key theoretical perspective motivating our approach comes from the information-theoretic interpretation of Shannon entropy. Given a categorical distribution $p = (p_1, p_2, \ldots, p_C)$, the Shannon entropy is defined as $H(p) = -\sum_{i=1}^{C} p_i \log p_i$. While $H$ is commonly viewed as a measure of uncertainty, it also directly encodes the notion of an effective number of categories, defined as $N_{\text{eff}}(p) = \exp(H(p))$. This quantity represents the number of outcomes in a uniform distribution that would yield the same level of uncertainty as $p$. For instance,

$$p = [0.8, 0.1, 0.1] \quad \Rightarrow \quad H(p) \approx 0.639, \quad N_{\text{eff}} \approx 1.89,$$

whereas

$$p = [0.3, 0.3, 0.4] \quad \Rightarrow \quad H(p) \approx 1.089, \quad N_{\text{eff}} \approx 2.97.$$

Although both are distributions over three categories, the first case exhibits a stronger bias towards a single outcome, hence the small probabilities $0.1$ are more likely to represent noise or modeling errors rather than genuine equiprobable alternatives, whereas the second distribution is closer to true ambiguity among three plausible options. This principle is precisely the motivation behind the notion of effective rank, introduced in (Roy & Vetterli, 2007). Analogous to the effective number of categories, effective rank measures the *effective dimensionality* of the matrix, which in our method corresponds to the number of equivalent semantic categories represented by the embedding vectors. As shown in Figure 2, we can intuitively see how the effective rank continuously measures the degree of divergence of matrix vectors.

Moreover, the effective rank employs the singular value distribution to calculate the Shannon entropy because the singular values of a matrix inherently capture information about the direction and magnitude distribution of its internal vectors. The more uniform the singular value distribution, the more dispersed the linearly independent vector groups within the matrix are; conversely, the more concentrated they are:

- If all column vectors share the same direction, then $A$ has rank one, and there is only a single non-zero singular value, thus $H = 0$, $\exp(H) = 1$, implying complete certainty.

- If all column vectors have the same length and are pairwise orthogonal, then all singular values are equal. In this case, $H = -\Sigma_{i=1}^{m} \frac{1}{m} \ln(\frac{1}{m}) = \ln m$, $\exp(H) = m$, which corresponds to maximal uncertainty.

We prove that the effective rank of a matrix is always less than or equal to its true rank, and the equality holds if and only if the two conditions above are satisfied. Let $A$ be a matrix with singular values $\sigma_1, \sigma_2, \ldots, \sigma_n$ where $n$ is the minimum dimension of $A$. The normalized singular values are defined as $p_i = \sigma_i / \|\sigma\|_1$, where $\|\sigma\|_1 = \sum_{i=1}^{n} \sigma_i$. The entropy of $A$ is given by $H(A) = -\sum_{i=1}^{n} p_i \log p_i$ (with $0 \log 0 = 0$), and the entropy-based effective rank is $\text{erank}(A) = \exp(H(A))$. The true rank is $r = \text{rank}(A)$, the number of non-zero singular values.

To prove $\text{erank}(A) \leq r$, we apply Jensen's inequality to the concave function $\log x$:

$$H(A) = \sum_{i:p_i>0} p_i \log\left(\frac{1}{p_i}\right) \leq \log\left(\sum_{i:p_i>0} p_i \cdot \frac{1}{p_i}\right) = \log r \tag{8}$$

where the sum is over the $r$ indices with $p_i > 0$. Thus, $\text{erank}(A) = \exp(H(A)) \leq \exp(\log r) = r$.

Equality holds if and only if all non-zero singular values are equal, since $\log x$ is strictly concave and equality in Jensen's inequality requires that all $1/p_i$ are equal for $p_i > 0$, meaning $p_i = 1/r$ for each non-zero singular value.

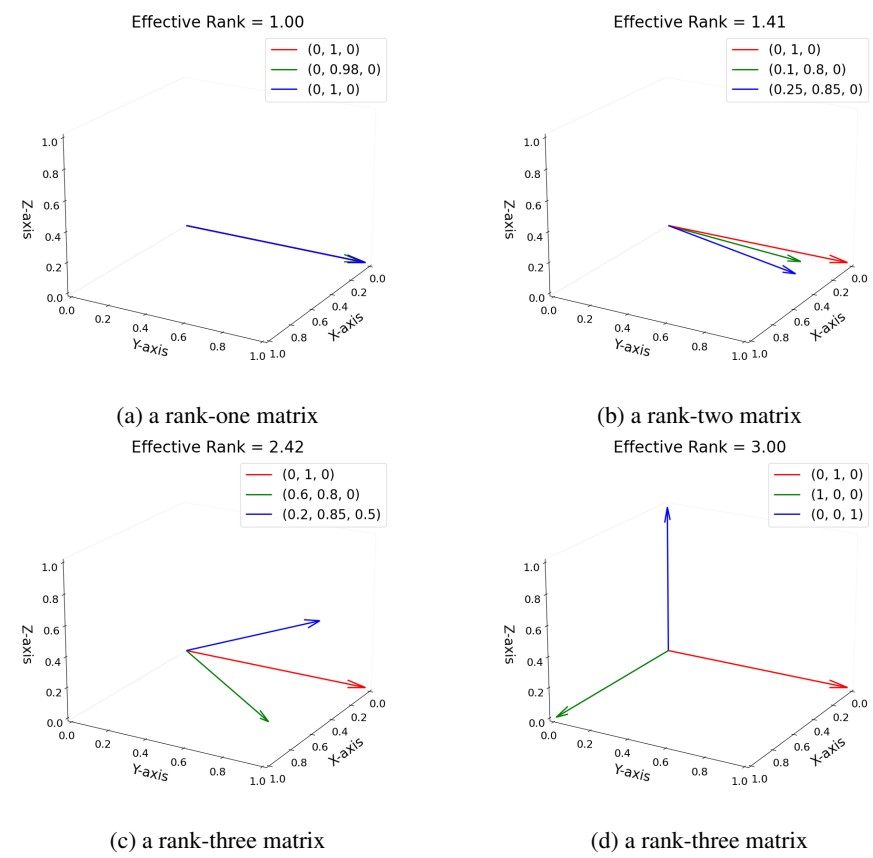

(a) a rank-one matrix

(b) a rank-two matrix

(c) a rank-three matrix

(d) a rank-three matrix

Figure 2: Visualization of Effective Ranks

## C  PROOFS AND DISCUSSION

This appendix provides the detailed proofs for the lemmas and proposition stated in the main text.

### C.1  PROOF OF LEMMA 1

(Variance Propagation with Expansion Property) For a fixed parameter $\theta$, the variance of the hidden state $h_t$ can be bounded from below recursively. Assuming the transformation $f$ exhibits representational expansion in deep Transformers, the conditional variance satisfies:

$$\mathbb{E}_\theta[\text{Var}(h_t|\theta)] \geq \mathbb{E}_\theta \mathbb{E}_{h_{t-1}} \left[ |J_y(h_{t-1})|_F^2 \cdot \text{Var}(y_t|h_{t-1};\theta) \right] + \Delta_{\text{nonlin}} \tag{9}$$

where $J_y = \frac{\partial f}{\partial y}$ is the Jacobian of the transformation with regard to the input token embedding, $|\cdot|_F$ denotes the Frobenius norm, and $\Delta$nonlin is a non-negative term capturing higher-order effects that typically amplify variance.

*Proof.* We analyze the evolution of the variance through the deterministic function $h_t = f(h_{t-1}, y_t; \theta)$. Our goal is to find a lower bound for $\text{Var}(h_t|\theta)$.

We begin by applying the law of total variance conditional on $\theta$:

$$\text{Var}(h_t|\theta) = \mathbb{E}_{h_{t-1}}[\text{Var}(h_t|h_{t-1}, \theta)] + \text{Var}_{h_{t-1}}(\mathbb{E}[h_t|h_{t-1}, \theta]) \tag{10}$$

We now analyze each term separately.

1. Analysis of $\mathbb{E}_{h_{t-1}}[\text{Var}(h_t|h_{t-1}, \theta)]$:

Given $h_{t-1}$, the randomness in $h_t$ comes solely from $y_t$. Let us define the conditional mean of the next token:

$$\mu_y(h_{t-1}) = \mathbb{E}[y_t|h_{t-1};\theta]$$

We perform a first-order Taylor expansion of the function $f$ around the point $(h_{t-1}, \mu_y(h_{t-1}))$:

$$h_t = f(h_{t-1}, y_t;\theta) = f(h_{t-1}, \mu_y;\theta) + J_y(h_{t-1}) \cdot (y_t - \mu_y) + R(h_{t-1}, y_t) \tag{11}$$

where $J_y(h_{t-1}) = \left.\frac{\partial f(h_{t-1}, y;\theta)}{\partial y}\right|_{y=\mu_y}$ is the Jacobian matrix, $R(h_{t-1}, y_t)$ is the Taylor remainder term, which encapsulates all higher-order derivatives.

Taking the conditional expectation $\mathbb{E}[\cdot|h_{t-1}, \theta]$ of Eq. 11:

$$\mathbb{E}[h_t|h_{t-1}, \theta] = f(h_{t-1}, \mu_y;\theta) + J_y(h_{t-1}) \cdot \underbrace{\mathbb{E}[(y_t - \mu_y)|h_{t-1}, \theta]}_{=0} + \mathbb{E}[R|h_{t-1}, \theta]$$

$$= f(h_{t-1}, \mu_y;\theta) + \mathbb{E}[R|h_{t-1}, \theta]$$

The conditional variance $\text{Var}(h_t|h_{t-1}, \theta)$ is the variance of the right-hand side of Eq. 11 around this conditional mean. Neglecting the covariance between the linear and remainder terms, we can approximate:

$$\text{Var}(h_t|h_{t-1}, \theta) \gtrsim \text{Var}\left(J_y(h_{t-1})(y_t - \mu_y) \mid h_{t-1}, \theta\right) + \text{Var}(R|h_{t-1}, \theta) \tag{12}$$

The first term is the variance of the linear approximation. Assuming $J_y$ is approximately constant over the variation of $y_t$ (because the internal transformations of well-trained LLMs often exhibit a certain degree of consistency in generating different sequences), this variance can be expressed as:

$$\text{Var}\left(J_y(y_t - \mu_y) \mid h_{t-1}, \theta\right) = J_y \cdot \text{Var}(y_t|h_{t-1};\theta) \cdot J_y^\top$$

The total variance of this linear term is the trace of this covariance matrix:

$$\text{Tr}\left(J_y \cdot \text{Var}(y_t|h_{t-1};\theta) \cdot J_y^\top\right) = \text{Tr}\left(J_y^\top J_y \cdot \text{Var}(y_t|h_{t-1};\theta)\right) = |J_y|_F^2 \cdot \text{Var}(y_t|h_{t-1};\theta)$$

The second term in Eq. 12, $\text{Var}(R|h_{t-1}, \theta)$, is the variance of the remainder. For common modern activation functions which are smooth and exhibit local linearity, and under the influence of layer normalization which constrains inputs, the remainder term $R$ is typically small. We denote this small contribution as $\delta_{\text{nonlin}}(h_{t-1})$. Crucially, for expansive transformations (Sanford et al., 2023; Wang et al., 2022), the higher-order terms often amplify variance rather than suppress it. Moreover, $J_y$ itself contains a large amount of parameter information. Therefore, the Frobenius norm $|J_y(h_{t-1})|_F$, which represents the square root of the sum of the squares of the matrix elements, is often much greater than 1.

Thus, we can write:

$$\text{Var}(h_t|h_{t-1}, \theta) \gtrsim |J_y(h_{t-1})|_F^2 \cdot \text{Var}(y_t|ht-1;\theta) + \delta_{\text{nonlin}}(h_{t-1}) \tag{13}$$

Taking the expectation of Eq. 13 with respect to $h_{t-1}$ and $\theta$ gives the first term of our final bound:

$$\mathbb{E}_\theta \mathbb{E}_{h_{t-1}}[\text{Var}(h_t|h_{t-1}, \theta)] \geq \mathbb{E}_\theta \mathbb{E}_{h_{t-1}}\left[|J_y(h_{t-1})|_F^2 \cdot \text{Var}(y_t|h_{t-1};\theta)\right] + \Delta_{\text{nonlin}}^{(1)} \tag{14}$$

where $\Delta_{\text{nonlin}}^{(1)} = \mathbb{E}_\theta \mathbb{E}_{h_{t-1}}[\delta_{\text{nonlin}}(h_{t-1})]$. $\qquad\square$

## C.2 Proof of Lemma 2

[Epistemic Uncertainty Bound] The epistemic variance can be bounded by:

$$\text{Var}_\theta(\mathbb{E}[h_t|\theta]) \leq |G_t|_F^2 \cdot \text{Tr}(\Sigma\theta) + \epsilon_t$$

*Proof.* Let $\mu_\theta = \mathbb{E}[\theta]$ be the mean of the parameter posterior distribution and $\delta = \theta - \mu_\theta$ be the zero-mean perturbation vector with covariance $\Sigma_\theta = \mathbb{E}[\delta\delta^\top]$.

Consider the expected hidden state $\mu_h(\theta) = \mathbb{E}[h_t|\theta]$ as a function of the parameters. We perform a first-order Taylor expansion around $\mu_\theta$:

$$\mu_h(\theta) = \mu_h(\mu_\theta + \delta) \approx \mu_h(\mu_\theta) + G_t \cdot \delta \tag{15}$$

where $G_t = \frac{\partial \mu_h(\theta)}{\partial \theta}\Big|_{\theta = \mu\theta}$, $G_t$ is the Jacobian matrix describing the sensitivity of the expected hidden state to parameter changes. The variance of $\mu_h(\theta)$ is then:

$$\text{Var}_\theta(\mathbb{E}[h_t|\theta]) = \text{Var}\theta(\mu_h(\theta)) \approx \text{Var}_\theta(G_t \cdot \delta) = \mathbb{E}[(G_t\delta)(G_t\delta)^\top] = G_t\mathbb{E}[\delta\delta^\top]G_t^\top = G_t\Sigma_\theta G_t^\top \tag{16}$$

To find the total variance, we take the trace of this covariance matrix:

$$\text{Tr}(\text{Var}_\theta(\mu_h(\theta))) \approx \text{Tr}(G_t\Sigma_\theta G_t^\top) = \text{Tr}(G_t^\top G_t\Sigma_\theta) \tag{17}$$

Applying the Cauchy-Schwarz inequality for the trace (von Neumann's trace inequality), we get:

$$\text{Tr}(G_t^\top G_t\Sigma_\theta) \leq \sqrt{\text{Tr}((G_t^\top G_t)^2) \cdot \text{Tr}(\Sigma_\theta^2)} \quad \text{(a)} \quad \leq \text{Tr}(G_t^\top G_t) \cdot \text{Tr}(\Sigma_\theta) \quad \text{(b)} \tag{18}$$

Inequality (a) is the Cauchy-Schwarz application. Inequality (b) holds because: $\text{Tr}(A^2) \leq (\text{Tr}(A))^2$ for a positive semidefinite matrix $A$. Noting that $\text{Tr}(G_t^\top G_t) = |G_t|_F^2$, we arrive at:

$$\text{Tr}(\text{Var}_\theta(\mu_h(\theta))) \lesssim |G_t|_F^2 \cdot \text{Tr}(\Sigma_\theta)$$

The term $\epsilon_t$ is introduced to account for the error in the first-order Taylor approximation, which includes the effects of higher-order derivatives. This error is typically small if the function $\mu_h(\theta)$ is approximately linear in $\theta$ in the region of high posterior probability, an assumption linked to a peaked posterior. □

## C.3 PROOF OF PROPOSITION 1

For an autoregressive language model exhibiting a sufficiently peaked parameter posterior and representational expansion over $t$ steps, the aleatoric uncertainty in its hidden representations may dominate the epistemic uncertainty:

$$\mathbb{E}_\theta[\text{Var}(h_t|\theta)] \gg \text{Var}_\theta(\mathbb{E}[h_t|\theta]) \tag{19}$$

*Proof.* Let $a_t = \mathbb{E}_\theta[\text{Var}(h_t|\theta)]$ (aleatoric) and $e_t = \text{Var}_\theta(\mathbb{E}[h_t|\theta])$ (epistemic).

From Lemma 1, the aleatoric term follows a recursive inequality:

$$a_t \gtrsim \mathbb{E}_\theta\mathbb{E}_{h_{t-1}}\left[|J_y(h_{t-1})|_F^2 \cdot \text{Var}(y_t|h_{t-1};\theta)\right]$$

Under the *representational expansion* assumption, the expected Jacobian norm $\mathbb{E}[|J_y|_F^2]$ is large ($\gg 1$). Furthermore, for most tokens in a generation, the predictive distribution $p(y_t|h_{t-1};\theta)$ has high entropy, meaning $\text{Var}(y_t|h_{t-1};\theta)$ is also significant. The product of these two large factors is a large positive quantity at each step $t$.

From Lemma 2, the epistemic term is bounded:

$$e_t \leq |G_t|_F^2 \cdot \text{Tr}(\Sigma_\theta) + \epsilon_t$$

Under the *peaked parameter posterior* assumption, $\text{Tr}(\Sigma_\theta)$ is small. The sensitivity $|G_t|_F^2$ of the expected hidden state to parameters depends on the model's stability. In a well-trained robust transformer with residual connections and layer normalization, the expected representations are often stable with respect to small parameter perturbations, restricting $|G_t|_F^2$ to a smaller range. The error term $\epsilon_t$ from the linear approximation is also small due to the peaked posterior. Consequently, $e_t$ remains bounded by a relatively small constant as $t$ increases.

Therefore, for each step $t$, the recursive accumulation of sampling noise through expansive transformations causes $a_t$ to become much larger than the bounded epistemic term $e_t$, leading to the stated dominance: $a_t \gg e_t$. □

## C.4 DISCUSSION OF LIMITATIONS

It is important to emphasize that the above analysis is heuristic rather than a strict universal proof. Several limitations apply. First, the variance decomposition in Eq. 4 is borrowed from Bayesian

deep learning for model outputs, but here it is applied to hidden states of autoregressive LLMs. This approach is heuristic and requires us to define the distribution function of LLM parameters in additional ways (although in practical applications, the parameters of LLMs are often given). Second, Lemma 1 relies on first-order Taylor expansions; the neglected higher-order term $\Delta_{\text{nonlin}}$ may be non-negligible in some cases. Finally, the bound on epistemic uncertainty in Lemma 2 assumes a concentrated posterior distribution and smooth sensitivity to parameters. While reasonable for large pretrained models, this may not hold for small or under-regularized ones. Therefore, the dominance of aleatoric over epistemic uncertainty in Proposition 1 should be viewed as a qualitative tendency under certain conditions, not a universal law. We highlight these caveats to clarify that our theoretical motivation is primarily heuristic, aiming to explain why single-pass-based hallucination is often unreliable and provide interpretable insights for future works.

## D  MORE INFORMATION ABOUT THE BASELINES

### D.1  COMPARISON OF EFFECTIVE RANK AND EIGENSCORE IN THEIR MATHEMATICAL ESSENCE

The EigenScore metric is computed through a structured procedure that leverages embedding vectors in the internal states of LLMs. Given an input query, we generate $K$ responses and extract their corresponding sentence embeddings, typically obtained from the last token of the middle layer, forming the embedding matrix $\mathbf{Z} \in \mathbb{R}^{d \times K}$. The centered covariance matrix is computed as $\mathbf{\Sigma} = \mathbf{Z}^\top \mathbf{J}_d \mathbf{Z}$, where $\mathbf{J}_d = \mathbf{I}_d - \frac{1}{d}\mathbf{1}_d\mathbf{1}_d^\top$ represents the centering matrix. After applying regularization $\mathbf{\Sigma}_{\text{reg}} = \mathbf{\Sigma} + \alpha\mathbf{I}_K$ to ensure full rank, we calculate the EigenScore as:

$$E = \frac{1}{K}\log\det(\mathbf{\Sigma}_{\text{reg}}) = \frac{1}{K}\sum_{i=1}^{K}\log(\lambda_i) \tag{20}$$

where $\lambda_i$ denotes the eigenvalues of $\mathbf{\Sigma}_{\text{reg}}$.

This formulation differs fundamentally from the effective rank approach based on Shannon entropy of the singular value spectrum. While EigenScore operates on eigenvalue decomposition of the covariance matrix, the effective rank method relies on singular value decomposition of the original embedding matrix $\mathbf{Z}$. Specifically, after computing singular values $\sigma_1, \sigma_2, \ldots, \sigma_r$ and normalizing them to form a probability distribution $p_i = \sigma_i / \sum_{j=1}^{r}\sigma_j$, the effective rank is derived as:

$$H = -\sum_{i=1}^{r} p_i \log p_i \tag{21}$$

Although the singular values of the matrix $Z$ are exactly the eigenvalues of $Z^\top Z$, and the eigenvalues of the covariance matrix are equivalent to the singular values of the centered $Z$, there are still some differences between the two (especially when the hidden-layer vectors have large norms but are close to each other). In addition, the two methods handle the eigenvalue or singular-value spectrum quite differently. Eigenscore requires the determinant of the covariance matrix (equivalently, the product of its eigenvalues), which demands that the covariance matrix be non-singular; therefore, a correction term must be added to ensure the determinant is non-zero. In contrast, the effective-rank method is more flexible: since $x \ln x \to 0$ as $x \to 0$, we can simply ignore singular values that are zero or very close to zero, avoiding the need to introduce such correction terms.

Moreover, Eigenscore and effective rank also differ significantly in their practical meanings. Eigenscore is an approximation of the internal differential entropy of an LLM, whereas effective rank reflects the equivalent number of semantic categories represented by the sampled responses. As shown in the case study in Appendix F, the computed eigenscores are all negative, while the effective ranks are positive real numbers between 1 and 10 (number of generations), and they can intuitively capture the semantic dispersion among different responses.

## D.2 SEMANTIC NEAREST-NEIGHBOR ENTROPY (SNNE)

As a simple yet effective uncertainty quantification method, **Semantic Nearest-Neighbor Entropy (SNNE)** is designed to overcome the limitations of Semantic Entropy in long-generation settings. SE clusters sampled answers using NLI-based entailment and computes entropy over cluster probabilities, but it fails to capture two crucial signals: *intra-cluster similarity* (the spread of answers within each semantic group) and *inter-cluster similarity* (the distance between different groups). As modern LLMs increasingly generate longer one-sentence outputs, these missing components tend to be amplified as the generated token length increases, which undermines the accuracy of hallucination detection.

SNNE addresses this by replacing explicit clustering with an aggregation of pairwise semantic similarities among sampled answers:

$$\text{SNNE}(q) = -\frac{1}{n} \sum_{i=1}^{n} \log \sum_{j=1}^{n} \exp\left( \frac{f(a_i, a_j \mid q)}{\tau} \right),$$ (22)

where $f$ is a semantic similarity function (e.g., ROUGE-L, entailment, or embedding cosine similarity). This smoothly incorporates both intra- and inter-cluster relations while remaining robust to outliers. The authors also introduce a white-box variant (WSNNE) that weights each answer by its normalized sequence probability. They further show theoretically that (W)SNNE generalizes SE and DSE under specific choices of $f$.

## D.3 SHIFTING ATTENTION TO RELEVANCE (SAR)

There is a key limitation in existing uncertainty quantification methods such as Predictive Entropy for free-form LLMs: *generative inequality*, the fact that tokens and sentences contribute unequally to the underlying semantics, yet are weighted equally when computing uncertainty. Empirical analysis shows that semantically irrelevant tokens or sentences often dominate the total uncertainty, significantly degrading hallucination detection.

To address this issue, the authors propose **SAR** method, which explicitly reweights uncertainty estimation using semantic relevance. At the token level, SAR measures how much removing each token changes the meaning of the generated sentence and scales its entropy contribution accordingly. At the sentence level, SAR boosts the generative probability of sentences that are semantically consistent with other sampled generations. The two components are combined to produce a relevance-aware predictive entropy that emphasizes meaningful semantic units and suppresses noise.

$$\text{SAR}(S) = \frac{1}{K} \sum_{j=1}^{K} -\log \left( p'(s_j) + \frac{1}{t} \sum_{k \neq j} g(s_j, s_k)\, p'(s_k) \right).$$ (23)

where $s_j$ denotes the $j$-th sampled generation, $K$ denotes the number of sampled generations, $p'(s_j)$ denotes the token-shifted probability of $s_j$, $g(s_j, s_k)$ denotes the semantic similarity between sentences $s_j$ and $s_k$, $t$ denotes the temperature controlling the strength of sentence-level shifting, and $p'(s_k)$ denotes the token-shifted probability of another sampled generation.

# E ADDITIONAL EXPERIMENT RESULTS

In this section, we provide complete experiment results from our ablation study on the number of generations, hidden-layer selection strategy, and temperature. It offers a finer-grained view of how the information encoded in hidden-layer representations interacts with these hyperparameters.

## E.1 HIDDEN LAYER SELECTION STRATEGY

Based on the AUROC results obtained using the Effective Rank method across different hidden layers of the Llama-2-7b-chat model, as illustrated in Figure 3, the optimal layer selection strategy for hallucination detection varies depending on the dataset type. For fact-based short-answer questions like TriviaQA, choosing any layer in the middle to late layers does not make a significant difference, where semantic representations balance surface-level facts and deeper context. In contrast, for news

summarization tasks such as CNN Dailymail, AUROC is not very stable in the middle and later layers, but it is still safer to choose the last few layers, as these layers capture high-level semantic coherence essential for summarizing content. For mathematical reasoning in MATH-500, AUROC has an increasing trend with forward propagation of layers, so selecting the last few layers is best, which handle complex logical structures. Overall, these findings suggest that Effective Rank-based hallucination detection should prioritize middle to late hidden layers, avoiding very early layers, to maximize AUROC across diverse tasks.

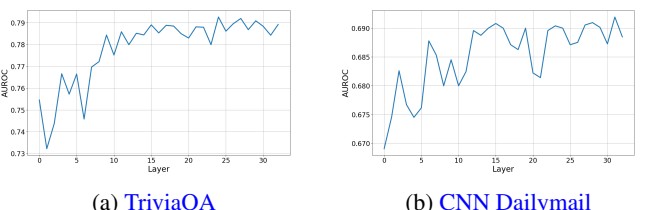

(a) TriviaQA        (b) CNN Dailymail       (c) MATH-500

Figure 3: The AUROC results of the three datasets using the Effective Rank method on the Llama-2-7b-chat model, each result is computed by selecting one hidden layer at a time. The layer indices start from 0 and follow the forward-propagation order of the hidden layers.

### E.2 Model-Specific Patterns in Hallucination Detection Performance

The ablation studies demonstrate distinct optimization strategies for hallucination detection across different model architectures. For smaller models like Llama-2-7b-chat, a single middle-layer extraction approach consistently delivers stable and superior performance, indicating that focused semantic representations in intermediate layers are most effective. In contrast, larger models such as Llama-2-13b-chat exhibit greater variability, requiring adaptive layer selection between final and middle layers depending on the number of generations, which suggests increased complexity in feature distribution. The Mistral model achieves optimal results with M5 strategy, highlighting the robustness of M5 in capturing semantic coherence. Temperature sensitivity analysis further reveals that moderate sampling diversity enhances detection reliability, while extreme values lead to significant performance degradation. Overall, the proposed layer-wise hidden state methods consistently match or surpass established baselines, validating the importance of architectural considerations in hallucination detection.

## F Case Studies and Further Discussions

Below, we provide several specific examples from the main experiment and a deeper analysis of the effectiveness of our method.

### F.1 TriviaQA Dataset on Llama-2-7b-chat

> **Question:** Who was the first man sent into space, in 1961?
> **Correct Answer:** ['gagarin']
> **LLM's Answer:** yuri gagarin
> **Accuracy:** 1.0
> **Sampled Responses:** ['yuri gagarin', 'yuri gagarin', 'yuri gagarin', 'yuri gagarin', 'yuri gagarin', 'yuri gagarin', 'yuri gagarin', 'yuri gagarin', 'yuri gagarin']
> **Singular Values:** [76.35235595703125, 2.6761877219491637e-14, 1.1108339471383637e-15, 2.146262724714404e-30, 3.9155441760212304e-31, 0.0, 0.0, 0.0, 0.0, 0.0]
> **Effective Rank:** 1.0000000000000002
> **Eigenscore:** -61.72965268471336

Table 6: Ablation studies on Llama-2-7b-chat, where N refers to the number of generations, M1 refers to extracting the exact middle hidden layer for each answer in the ER method, M5 refers to extracting the middle five layers, L1 refers to extracting the last layer, and L5 refers to extracting the last five layers, ES denotes Eigenscore, PF denotes P_False, DSE denotes Discrete Semantic Entropy, LNE denotes Length-Normalized Entropy, and SE denotes Semantic Entropy. All numbers in the table are AUROC scores; values closer to 1 indicate stronger hallucination-detection ability

| N | Dataset | M1 | M5 | L1 | L5 | ES | PF | DSE | LNE | SE |
|---|---------|-----|-----|-----|-----|-----|-----|-----|-----|-----|
| 10 | TriviaQA | **0.7877** | 0.7700 | 0.7809 | 0.7629 | 0.7802 | 0.6625 | 0.7758 | 0.6947 | 0.7750 |
| | SQuAD | 0.7212 | **0.7217** | 0.7191 | 0.7190 | 0.7199 | 0.6594 | 0.7172 | 0.6505 | 0.7181 |
| | BioASQ | 0.8447 | 0.8461 | **0.8481** | 0.8472 | 0.8433 | 0.7321 | 0.8437 | 0.4703 | 0.8425 |
| | NQ | 0.7029 | 0.7038 | 0.7036 | 0.7049 | **0.7056** | 0.6584 | 0.6984 | 0.6495 | 0.7001 |
| | Average | **0.7641** | 0.7604 | 0.7629 | 0.7585 | 0.7623 | 0.6781 | 0.7588 | 0.6163 | 0.7589 |
| 15 | TriviaQA | 0.7862 | **0.7902** | 0.787 | 0.7807 | 0.7825 | 0.6579 | 0.7813 | 0.7211 | 0.7768 |
| | SQuAD | **0.7407** | **0.7407** | 0.7391 | 0.7395 | 0.7391 | 0.7148 | 0.7349 | 0.6070 | 0.7381 |
| | BioASQ | **0.8715** | 0.8690 | 0.8684 | 0.8678 | 0.8682 | 0.7496 | 0.8648 | 0.4748 | 0.8615 |
| | NQ | 0.7182 | 0.7166 | 0.7174 | 0.7188 | **0.7206** | 0.6598 | 0.7148 | 0.6823 | 0.7157 |
| | Average | **0.7792** | 0.7791 | 0.7780 | 0.7767 | 0.7776 | 0.6955 | 0.7740 | 0.6213 | 0.7730 |
| 20 | TriviaQA | **0.7786** | 0.7780 | 0.7729 | 0.7702 | 0.7743 | 0.6636 | 0.7671 | 0.7180 | 0.7698 |
| | SQuAD | 0.7596 | 0.7604 | 0.7599 | **0.7610** | 0.7579 | 0.6791 | 0.7588 | 0.6583 | 0.7581 |
| | BioASQ | **0.8645** | 0.8625 | 0.8638 | 0.8620 | 0.8628 | 0.7517 | 0.8564 | 0.4570 | 0.8585 |
| | NQ | 0.7277 | **0.7292** | 0.7274 | 0.7278 | 0.7284 | 0.6551 | 0.7243 | 0.6599 | 0.7256 |
| | Average | **0.7826** | 0.7825 | 0.7810 | 0.7803 | 0.7809 | 0.6874 | 0.7767 | 0.6233 | 0.7780 |

Table 7: Ablation studies on Llama-2-13b-chat about the number of generations and hidden layer vector extraction strategy

| N | Dataset | M1 | M5 | L1 | L5 | ES | PF | DSE | LNE | SE |
|---|---------|-----|-----|-----|-----|-----|-----|-----|-----|-----|
| 10 | TriviaQA | 0.7407 | **0.7516** | 0.7302 | 0.7495 | 0.7342 | 0.7379 | 0.7331 | 0.6930 | 0.7390 |
| | SQuAD | 0.7259 | 0.7273 | 0.7241 | 0.7228 | 0.7239 | 0.6945 | **0.7381** | 0.6886 | 0.7350 |
| | BioASQ | 0.8234 | 0.8277 | 0.8318 | **0.8324** | 0.8215 | 0.7980 | 0.7951 | 0.5610 | 0.7988 |
| | NQ | 0.7284 | 0.7290 | 0.7286 | **0.7352** | 0.7270 | 0.6841 | 0.7234 | 0.6866 | 0.7258 |
| | Average | 0.7546 | 0.7589 | 0.7537 | **0.7600** | 0.7517 | 0.7286 | 0.7474 | 0.6573 | 0.7497 |
| 15 | TriviaQA | 0.7763 | 0.7672 | 0.7681 | **0.7781** | 0.7654 | 0.7249 | 0.7590 | 0.6446 | 0.7577 |
| | SQuAD | 0.7289 | 0.7289 | 0.7271 | 0.7285 | 0.7264 | 0.6747 | **0.7393** | 0.6823 | 0.7379 |
| | BioASQ | **0.8358** | 0.8313 | 0.8341 | 0.8313 | 0.8328 | 0.7490 | 0.8224 | 0.4874 | 0.8148 |
| | NQ | 0.7015 | 0.6996 | 0.7045 | **0.7054** | 0.7038 | 0.6902 | 0.6999 | 0.6452 | 0.6936 |
| | Average | **0.7606** | 0.7560 | 0.7577 | 0.7601 | 0.7571 | 0.7097 | 0.7567 | 0.6149 | 0.7523 |
| 20 | TriviaQA | 0.7439 | **0.7553** | 0.7446 | 0.7339 | 0.7246 | 0.7294 | 0.7413 | 0.6953 | 0.7429 |
| | SQuAD | 0.7331 | 0.7306 | 0.7293 | 0.7291 | 0.7299 | 0.6589 | **0.7382** | 0.6874 | 0.7360 |
| | BioASQ | 0.8409 | 0.8454 | 0.8411 | **0.8461** | 0.8413 | 0.7864 | 0.8279 | 0.5939 | 0.8299 |
| | NQ | 0.7178 | 0.7155 | **0.7193** | 0.7185 | 0.7175 | 0.6428 | 0.7149 | 0.6880 | 0.7120 |
| | Average | 0.7589 | **0.7617** | 0.7586 | 0.7569 | 0.7533 | 0.7044 | 0.7556 | 0.6662 | 0.7552 |

---

**Question:** September 9, 1969 saw what made an official language of Canada?
**Correct Answer:** ['french']
**LLM's Answer:** french
**Accuracy:** 1.0
**Sampled Responses:** ['Quebec', 'french', 'french', 'Quebec', 'constitution', 'Quebec', 'french', 'french', 'french']
**Singular Values:** [60.25969314575195, 41.63536071777344, 24.346080780029297, 1.6957731741170864e-14, 5.443248018391789e-15, 1.198083634661477e-15, 8.686304874642721e-16, 1.4343240353264575e-16, 8.151879937535818e-32, 2.8225723498131095e-32]
**Effective Rank:** 2.818618681563689
**Eigenscore:** -51.25870849393463

Table 8: Ablation studies on Mistral-7B-v0.1 about the number of generations and hidden layer vector extraction strategy

| N | Dataset | M1 | M5 | L1 | L5 | ES | PF | DSE | LNE | SE |
|---|---------|----|----|----|----|----|----|-----|-----|-----|
| 10 | TriviaQA | 0.7634 | 0.7650 | **0.7651** | 0.7637 | 0.7579 | 0.7239 | 0.7575 | 0.6519 | 0.7553 |
| | SQuAD | 0.7208 | 0.7144 | 0.7073 | 0.7080 | 0.7120 | 0.6333 | 0.7227 | 0.6223 | **0.7238** |
| | BioASQ | 0.8563 | **0.8584** | 0.8506 | 0.8475 | 0.8513 | 0.7173 | 0.8507 | 0.5955 | 0.8513 |
| | NQ | 0.7658 | **0.7696** | 0.7614 | 0.7611 | 0.7627 | 0.7523 | 0.7678 | 0.6770 | 0.7662 |
| | Average | 0.7766 | **0.7769** | 0.7711 | 0.7701 | 0.7710 | 0.7067 | 0.7747 | 0.6367 | 0.7742 |
| 15 | TriviaQA | 0.7727 | **0.7767** | 0.7727 | 0.7720 | 0.7706 | 0.7265 | 0.7660 | 0.7058 | 0.7676 |
| | SQuAD | 0.6952 | 0.7064 | 0.6936 | 0.6907 | 0.6866 | 0.6047 | **0.7041** | 0.6529 | 0.6991 |
| | BioASQ | **0.8586** | 0.8548 | 0.8539 | 0.8557 | 0.8557 | 0.6866 | 0.8551 | 0.6129 | 0.8560 |
| | NQ | 0.7859 | 0.7861 | 0.7886 | **0.7896** | 0.7826 | 0.7717 | 0.7826 | 0.6975 | 0.7828 |
| | Average | 0.7781 | **0.7810** | 0.7772 | 0.7770 | 0.7739 | 0.6974 | 0.7770 | 0.6673 | 0.7764 |
| 20 | TriviaQA | 0.7677 | 0.7672 | 0.7599 | **0.7688** | 0.7644 | 0.7663 | 0.7569 | 0.6861 | 0.7623 |
| | SQuAD | 0.7433 | 0.7480 | 0.7413 | 0.7384 | 0.7371 | 0.5995 | **0.7559** | 0.6782 | 0.7533 |
| | BioASQ | **0.8670** | 0.8662 | 0.8630 | 0.8631 | 0.8641 | 0.7594 | 0.8617 | 0.5854 | 0.8645 |
| | NQ | **0.7754** | 0.7749 | 0.7723 | 0.7724 | 0.7695 | 0.7709 | 0.7736 | 0.7046 | 0.7712 |
| | Average | 0.7884 | **0.7891** | 0.7841 | 0.7857 | 0.7838 | 0.7240 | 0.7870 | 0.6636 | 0.7878 |

Table 9: Ablation studies on Mistral-7B-v0.1 about temperature, where t denotes temperature

| t | Dataset | M1 | M5 | L1 | L5 | ES | PF | DSE | LNE | SE |
|---|---------|----|----|----|----|----|----|-----|-----|-----|
| 0.1 | TriviaQA | 0.6514 | 0.6660 | **0.6782** | 0.6503 | 0.6200 | 0.6695 | 0.6263 | 0.5391 | 0.6466 |
| | SQuAD | 0.6451 | **0.6466** | 0.6118 | 0.5971 | 0.6038 | 0.6413 | 0.6125 | 0.5506 | 0.6333 |
| | BioASQ | 0.7336 | 0.6834 | 0.7268 | 0.7416 | 0.6923 | **0.7720** | 0.7085 | 0.4401 | 0.7352 |
| | NQ | 0.6771 | 0.6665 | 0.6841 | 0.6664 | 0.6673 | **0.7308** | 0.6700 | 0.6231 | 0.6822 |
| | Average | 0.6768 | 0.6656 | 0.6752 | 0.6639 | 0.6459 | **0.7034** | 0.6543 | 0.5382 | 0.6743 |
| 0.5 | TriviaQA | 0.7438 | **0.7628** | 0.7518 | 0.7492 | 0.7350 | 0.6806 | 0.7280 | 0.5880 | 0.7154 |
| | SQuAD | 0.7364 | 0.7391 | 0.7392 | **0.7401** | 0.7330 | 0.6551 | 0.7308 | 0.6104 | 0.7318 |
| | BioASQ | 0.8432 | 0.8374 | 0.8449 | **0.8452** | 0.8378 | 0.7320 | 0.8355 | 0.5167 | 0.8293 |
| | NQ | 0.7679 | 0.7671 | **0.7724** | 0.7678 | 0.7641 | 0.7516 | 0.7643 | 0.6356 | 0.7629 |
| | Average | 0.7728 | 0.7766 | **0.7771** | 0.7756 | 0.7675 | 0.7048 | 0.7647 | 0.5877 | 0.7599 |
| 1.0 | TriviaQA | 0.7634 | 0.7650 | **0.7651** | 0.7637 | 0.7579 | 0.7239 | 0.7575 | 0.6519 | 0.7553 |
| | SQuAD | 0.7208 | 0.7144 | 0.7073 | 0.7080 | 0.7120 | 0.6333 | 0.7227 | 0.6223 | **0.7238** |
| | BioASQ | 0.8563 | **0.8584** | 0.8506 | 0.8475 | 0.8513 | 0.7173 | 0.8507 | 0.5955 | 0.8513 |
| | NQ | 0.7658 | **0.7696** | 0.7614 | 0.7611 | 0.7627 | 0.7523 | 0.7678 | 0.6770 | 0.7662 |
| | Average | 0.7766 | **0.7769** | 0.7711 | 0.7701 | 0.7710 | 0.7067 | 0.7747 | 0.6367 | 0.7742 |
| 2.0 | TriviaQA | 0.6679 | 0.6685 | 0.6651 | 0.6662 | 0.6545 | **0.7084** | 0.6643 | 0.6234 | 0.6666 |
| | SQuAD | **0.5754** | 0.5502 | 0.5593 | 0.5407 | 0.5698 | 0.5603 | 0.5213 | 0.5633 | 0.5592 |
| | BioASQ | 0.7280 | **0.7329** | 0.7106 | 0.7238 | 0.6998 | 0.6867 | 0.7243 | 0.5534 | 0.7244 |
| | NQ | 0.6034 | 0.6031 | 0.6118 | 0.6069 | 0.5788 | **0.7006** | 0.5996 | 0.5580 | 0.5906 |
| | Average | 0.6437 | 0.6387 | 0.6367 | 0.6344 | 0.6257 | **0.6640** | 0.6274 | 0.5745 | 0.6352 |

**Question:** In which part of the human body would you find the Sphenoid bone?
**Correct Answer:** ['skull']
**LLM's Answer:** brain
**Accuracy:** 0.0
**Sampled Responses:** ['head', 'brain', 'skull', 'skull', 'skull', 'head', 'brain', 'skull', 'skull']
**Singular Values:** [58.72682571411133, 34.81084442138672, 29.228200912475586, 7.700909307212667e-15, 7.15308397231237e-15, 2.269143032179147e-15, 3.835939579384829e-16, 3.338613852606579e-30, 1.4536969531393071e-31, 7.850294372204882e-33]
**Effective Rank:** 2.8627889069115606
**Eigenscore:** -51.30254888163167

## F.2 SQUAD DATASET ON MISTRAL-7B-V0.1

**Question:** Who did Frick remove from the police force?
**Correct Answer:** ['anyone he suspected of being a republican']
**LLM's Answer:** the corrupt
**Accuracy:** 0.0
**Sampled Responses:** ['the corrupt', 'the corrupt', 'the corrupt', 'the corrupt', 'the corrupt', 'the corrupt', 'the corrupt', 'the corrupt', 'the corrupt']
**Singular Values:** [1139.242431640625, 3.5298562575496184e-13, 1.458547028271827e-14, 3.75700543160097e-29, 3.410722249588845e-30, 2.802596928649634e-45, 1.401298464324817e-45, 0.0, 0.0, 0.0]
**Effective Rank:** 1.0000000000000004
**Eigenscore:** -64.89193565768241

---

**Question:** Who verbally attacked Steve Stone?
**Correct Answer:** ['Kent Mercker']
**LLM's Answer:** Steve Stone
**Accuracy:** 0.0
**Sampled Responses:** ['Joe Morgan', 'his boss', 'John Kruk', 'Steve Stone', 'Steve Stone', 'his boss', 'John Madden', 'his boss', 'his boss']
**Singular Values:** [1180.1435546875, 552.9315185546875, 154.78468322753906, 16.52665138244629, 6.953697204589844, 1.193859735463404e-13, 1.553780128377754e-14, 6.809435505831249e-16, 1.553959034971104e-17, 1.1361135973578757e-30]
**Effective Rank:** 2.513272471125542
**Eigenscore:** -61.32968780733988

---

**Question:** In 2002 what act granted full British citizenship to the citizens of the islands?
**Correct Answer:** ['British Overseas Territories Act 2002']
**LLM's Answer:** British Overseas Territories Act
**Accuracy:** 1.0
**Sampled Responses:** ['British Overseas Territories Act', 'British Overseas Territories Act', 'British Overseas Territories Act, 'British Overseas Territories Act', 'British Overseas Territories Act', 'British Nationality Act', 'British Overseas Territories Act', 'British Overseas Territories Act', 'British Overseas Territories Act']
**Singular Values:** [1153.8829345703125, 328.6450500488281, 7.30688702902868e-14, 2.118430780068855e-30, 1.2843073833195104e-33, 0.0, 0.0, 0.0, 0.0, 0.0]
**Effective Rank:** 1.6972759552279857
**Eigenscore:** -63.12688798174471

## F.3 BioASQ Dataset on Llama-2-13b-chat

**Question:** Which are the thyroid hormone analogs utilized in human studies?
**Correct Answer:** ['TRIAC']
**LLM's Answer:** Liothyronine (T3) and levothyroxine (T4)
**Accuracy:** 0.0
**Sampled Responses:** ['triiodothyronine (T3) and levothyroxine (T4)', 'Liothyronine (T3) and levothyroxine (T4)', 'Liothyronine (T3) and levothyroxine (T4)', 'Liothyronine (T3) and levothyroxine (T4)', 'Liothyronine (T3) and levothyroxine (T4)', 'Liothyronine (T3) and levothyroxine (T4)', 'liothyronine and levothyroxine', 'Liothyronine (T3) and levothyroxine (T4)', 'liothyronine and levothyroxine']
**Singular Values:** [218.68751525878906, 62.00069046020508, 15.469388961791992, 3.4699248902941024e-15, 3.3113897569630145e-15, 4.50245604514668e-31, 1.978491468987644e-31, 2.0052295279776766e-32, 0.0, 0.0]
**Effective Rank:** 2.0248367530554368
**Eigenscore:** -53.52278438073435

---

**Question:** Which genes are involved in patient response to warfarin?
**Correct Answer:** ['CYP2C9', 'VKORC1', 'ORM1', 'CYP4F2', 'EPHX1', 'CYP2C18', 'CYP2C19', 'CYP3A5', 'protein S', 'clotting factor V', 'PROC', 'GGCX']
**LLM's Answer:** VKORC1 and CYP2C9
**Accuracy:** 1.0
**Sampled Responses:** ['VKORC1 and CYP2C9', 'VKORC1 and CYP2C9', 'CYP2C9 and VKORC1', 'VKORC1 and CYP2C9', 'CYP2C9, VKORC1, and CYP3A4', 'CYP2C9, VKORC1, and CYP3A4', 'VKORC1 and CYP2C9', 'VKORC1 and CYP2C9', 'CYP2C9 and VKORC1']
**Singular Values:** [217.03753662109375, 100.93067932128906, 6.290593147277832, 4.6974177206424855e-15, 2.600824048733518e-15, 1.8994920306458578e-16, 1.1646331124343226e-16, 8.127036452178364e-32, 1.1876486869033015e-33, 0.0]
**Effective Rank:** 2.0309090152250144
**Eigenscore:** -53.3001540843245

---

**Question:** List symptoms of the IFAP syndrome.
**Correct Answer:** ['follicular ichthyosis', 'atrichia', 'photophobia']
**LLM's Answer:** intellectual disability, fibrosis, alopecia, and pigmentary changes.
**Accuracy:** 0.0
**Sampled Responses:** ['ichthyosis, facial dysmorphism, arthrogryposis, and pul', 'intellectual disability, fibrosis, alopecia, and pigmentary changes', 'Intellectual disability, FGFR3 mutation, cardiac defects, and pul', 'intellectual disability, fibrosis, ataxia, and pulmonary disease', 'ichthyosis, ataxia, pulmonary dysfunction, and immunodefic', 'intellectual disability, fibrosis, ataxia, and seizures', 'intellectual disability, fibrosis, alopecia, and pigmentation abnormalities', 'ichthyosis, facial dysmorphism, arthrogryposis, and pul', 'intellectual disability, fragile bones, anxiety, and pulmonary issues']
**Singular Values:** [122.30469512939453, 114.34846496582031, 82.5096206665039, 71.02500915527344, 61.05084991455078, 55.45585250854492, 55.206974029541016, 45.03531265258789, 5.794018268585205, 1.2034170623171584e-14]
**Effective Rank:** 7.804594598381604
**Eigenscore:** -22.963745969679774

## F.4 NQ Dataset on Mistral-7B-v0.1

> **Question:** last time east won nba all star game
> **Correct Answer:** ['2014']
> **LLM's Answer:** 2006
> **Accuracy:** 0.0
> **Sampled Responses:** ['2004', '2001', '2001', '2013', '2013', '2013', '2013', '2004', '2013']
> **Singular Values:** [1097.43212890625, 585.4111938476562, 355.2393798828125, 2.6302599906921387, 1.0910121140935217e-13, 7.080334175945183e-14, 6.622470177493692e-14, 9.815065427076775e-15, 4.458403975289517e-16, 3.7646645390169164e-30]
> **Effective Rank:** 2.731143238800789
> **Eigenscore:** -61.13279244771307

> **Question:** who was the temple of vesta built for?
> **Correct Answer:** ['Vesta']
> **LLM's Answer:** Vesta
> **Accuracy:** 1.0
> **Sampled Responses:** ['Vesta', 'the goddess vesta', 'Vesta', 'the goddess vesta', 'the goddess vesta', 'Vesta', 'Vesta', 'Vesta', 'Vesta']
> **Singular Values:** [1252.1671752929688, 372.3441772460938, 2.068291691343857e-13, 1.5302303423957858e-13, 1.352673246172607e-14, 2.020799899278259e-29, 9.934199431454563e-30, 4.8484518987236446e-30, 0.0, 0.0]
> **Effective Rank:** 1.7131135330533338
> **Eigenscore:** -62.060661269023285

> **Question:** what percentage of the world died in ww2?
> **Correct Answer:** ['about 3%']
> **LLM's Answer:** 3%
> **Accuracy:** 1.0
> **Sampled Responses:** ['3%', '3%', '3%', '3%', '3%', '3%', '3%', '3%', '3%']
> **Singular Values:** [1136.5360107421875, 3.5605334869695526e-13, 1.574521085343991e-14, 2.644209128265101e-29, 1.6524366075384805e-30, 2.802596928649634e-45, 0.0, 0.0, 0.0, 0.0]
> **Effective Rank:** 1.0000000000000004
> **Eigenscore:** -64.94262853905146

## F.5 Relationship between Hallucination and Uncertainty

From the above examples, we observe that hallucinations in LLMs are generally positively correlated with their uncertainty. However, counterexamples also exist, such as hallucinated answers with high confidence and correct answers with low confidence. We argue that this phenomenon mainly stems from the diverse causes of hallucination: (i) erroneous internal knowledge, (ii) insufficient internal knowledge and reasoning capability, (iii) excessive stochasticity, and (iv) lack of faithfulness during the generation process. Hallucinations caused by the latter three can often be detected via uncertainty estimation. In contrast, hallucinations due to erroneous internal knowledge are usually associated with low uncertainty, making them difficult to capture with uncertainty-based methods. Therefore, we recommend combining our uncertainty-based approach with knowledge editing and retrieval augmentation techniques to eliminate internal knowledge errors, thereby reducing highly consistent hallucinations and further enhancing AI trustworthiness.

## F.6 Reliability and Limitations of ROUGE-L for Hallucination Annotation

Based on the case analysis, we find that ROUGE-L is a reliable proxy for hallucination annotation in our experiments, particularly when the dataset contains abundant correct answers and these answers

are relatively short. This is because ROUGE-L, by computing the longest common subsequence, effectively checks whether key tokens from the ground-truth answer appear in the model output, similar to keyword-based grading in short-answer exams. In contrast, semantic vector similarity is more prone to misjudgment, as small variations such as additional predicates or modifiers can cause embeddings to be judged as semantically different. Moreover, some questions in our dataset have multiple semantically distinct correct answers, and LLMs may generate multiple answers simultaneously, making semantic similarity metrics less suitable. Another potentially effective annotation method is to determine whether the LLM output and the ground truth are in a bidirectional entailment relationship within context. However, this requires introducing an auxiliary NLI model, which greatly increases annotation complexity. In summary, ROUGE-L aligns well with the characteristics and needs of our datasets and experiments, providing satisfactory annotation reliability. Nevertheless, for more complex scenarios such as multi-turn QA or long-form generation, we recommend adopting more robust hallucination annotation methods.

### F.7 TIME COMPLEXITY ANALYSIS OF EFFECTIVE RANK

Although our method requires performing singular value decomposition (SVD) on a matrix composed of multiple high-dimensional vectors, modern libraries such as `numpy.linalg.svd()` make the computation highly efficient. The runtime analysis in the main text (Table 2) shows that the additional computation time of our Effective Rank method is negligible compared to the time required for answer generation. Moreover, its time cost is substantially lower than that of P_False (which introduces extra prompt-based verification) and Semantic Entropy (which requires semantic analysis). For an $m \times n$ matrix with $m < n$, the time complexity of SVD is $\mathcal{O}(m^2 n)$. In our main experiments, we compute matrices of size $10 \times 4096$ or $50 \times 4096$, where such complexity is entirely acceptable in practice.

### F.8 FAILURE CASE ANALYSIS

We have already discussed that although an LLM's uncertainty and hallucination are generally strongly correlated, there still exist many counterexamples, which represent the main limitation of using uncertainty to detect hallucinations. These counterexamples fall into two categories: hallucinated answers with low uncertainty, and correct answers with high uncertainty.

The former is illustrated by the first example in the case study on the SQuAD dataset in this section: for the question "Who did Frick remove from the police force?" the model consistently gives the wrong answer, "the corrupt," instead of the correct answer, "anyone he suspected of being a Republican." This occurs because, in the text-understanding and reasoning dataset, the LLM misinterprets the surrounding context, reflecting a limitation of the model's capability, which may be alleviated as LLMs continue to improve.

The latter is illustrated by the first example in the case study on TriviaQA: for the question "September 9, 1969 saw what made an official language of Canada?" the model gives the correct answer, "French," but the sampled responses also include incorrect answers such as "Quebec" and "constitution," which leads to a high measured uncertainty. From an optimistic perspective, such counterexamples are more beneficial than harmful for hallucination detection: even if a high-uncertainty answer happens to be correct, it is very likely that the LLM "guessed it right," which still reflects a weakness and provides a meaningful warning. Therefore, some uncertainty-based hallucination-detection works do not treat high-uncertainty correct answers as detection errors.

Overall, we observe that although some counterexamples appear in the 12 question–answer pairs in the case study, the overall trend still shows a clear positive correlation between uncertainty and hallucination, supporting the reliability of our method as well as other uncertainty-based baselines.

