# OpenReview forum: "Revisiting Hallucination Detection Through The Lens Of Effective Rank-based Uncertainty"
_ICLR.cc/2026/Conference — Submitted to ICLR 2026_

### Official Review · Reviewer_i66o · 2025-10-15

**Soundness:** 3
**Presentation:** 3
**Contribution:** 3
**Rating:** 8
**Confidence:** 4

**Summary:**

This paper introduced a novel uncertainty quantification approach for LLMs. The core idea of the approach is to compute the effective rank of multiple sampled generations. The authors conducted experiments on four datasets with three LLMs. The result showed that their proposed method outperforms other uncertainty quantification baselines on hallucination detection. The authors also provided a theoretical analysis of uncertainty quantification for LLMs, showing that the aleatoric uncertainty dominates the epistemic uncertainty and suggesting that their proposed approach effectively quantifies the aleatoric uncertainty.

**Strengths:**

1. The writing is clear and easy to follow.
2. The paper proposed a simple yet effective approach for uncertainty quantification, which would be useful for hallucination detection and improving the reliability of LLMs. In addition, the proposed approach is training-free, which would be easier to deploy in real-world setting.
3. The theoretical analysis provides further support for the proposed approach, making it more sound. In addition, it provides a justification of sampling-based uncertainty quantification, which will benefit the future research of uncertainty quantification for LLMs.

**Weaknesses:**

1. **Comparison to Eigenscore.** The proposed method is highly similar to Eigenscore [1]. Both aggregated the singular values of a matrix formed by the embedding of multiple generations. While I can see that the method proposed in this paper is simpler and theoretically grounded, it would be clearer if the authors could provide a detailed comparison to Eigenscore, showing the mathematical differences between these two approaches and the fundamental strength approach proposed in this paper.
2. **Baselines and LLMs.** There are other (more recent and powerful) uncertainty quantification baselines that were not included in this paper, such as [1, 2, 3, 4]. This paper could be strengthened more if the authors could show that their approach outperforms these additional baselines. In addition, the experiments are conducted on relatively outdated LLMs. It would be more convincing if the authors could include the performance of newer models, like Qwen3, Llama4, and Gemma3.
3. **Significance of the performance.** The performance of sampling-based uncertainty quantification depends on the stochastic nature of sampling. It would be great if the authors could report the variance/std of each approach, especially when their performance is close to Eigenscore.

[1]: INSIDE: LLMS’ INTERNAL STATES RETAIN THE POWER OF HALLUCINATION DETECTION (2024)

[2]: Enhancing Uncertainty-Based Hallucination Detection with Stronger Focus (2023)

[3]: Shifting Attention to Relevance: Towards the Predictive Uncertainty Quantification of Free-Form Large Language Models (2024)

[4]: HaloScope: Harnessing Unlabeled LLM Generations for Hallucination Detection (2024)

[5]: Steer LLM Latents for Hallucination Detection (2025)

**Questions:**

1. The current experiments were mainly conducted on short-form QA. I wonder whether ER can be applied to long-form QA, which is a more realistic setting.

---

> ### Author Response · Authors · 2025-11-21
> **Response to Reviewer i66o (Part 1/2)**
>
> Dear Reviewer i66o,
>
> Thank you for your valuable feedback and recognizing our work as clear, simple yet effective, and sound. In response to your concerns, we have carefully revised the manuscript and provide detailed responses below.
>
> ---
>
> **Q1: Comparison to Eigenscore.** The proposed method is highly similar to Eigenscore [1]. Both aggregated the singular values of a matrix formed by the embedding of multiple generations. While I can see that the method proposed in this paper is simpler and theoretically grounded, it would be clearer if the authors could provide a detailed comparison to Eigenscore, showing the mathematical differences between these two approaches and the fundamental strength approach proposed in this paper.
>
> **A1**: Thank you for your insightful comment. We acknowledge that our method shares some conceptual similarities with hidden state-based uncertainty quantification approaches, but we respectfully argue that the effective rank introduces a more interpretable and unified framework for this line of research. Specifically, effective rank conveys human-readable metrics (quantified as the numerical number of clusters) while it stands fundamentally apart from prior work such as Eigenscore, which only delivers uninterpretable numerical outputs for classification purposes. Beyond its intuitive interpretability, our method also outperforms existing baselines in both efficiency and effectiveness, offering comprehensive improvements over the state-of-the-art. Thus, we respectfully contend that conceptual overlap does not fully warrant the critique of insufficient novelty.
> Following your suggestion, we additionally compared our method with Eigenscore in **Appendix D** (as well as comparison with other baselines), particularly in their distinctions of mathematical formulations. We hope this detailed comparison can help readers better understand this difference.
>
> ---
>
> **Q2: Baselines and LLMs.** There are other (more recent and powerful) uncertainty quantification baselines that were not included in this paper, such as [2, 3, 4, 5]. This paper could be strengthened more if the authors could show that their approach outperforms these additional baselines. In addition, the experiments are conducted on relatively outdated LLMs. It would be more convincing if the authors could include the performance of newer models, like Qwen3, Llama4, and Gemma3.
>
> **A2**: Thank you for your kind suggestion and for sharing these excellent works with us. Following your suggestions, we have included SAR [3] as an additional baseline in Table 3 (copied below), since SAR **does not require external knowledge or extra module for detection (the same as our method, and we mainly compare with such methods)**, **while other methods need these additional entities**. Thus, due to time limitations, we have compared SAR in this version and cited other baselines in Section 2, and will compare all of them in the camera-ready version. Thanks again for the sharing!
>
> | Model | Task | Dataset | ER (Ours) | ES | PF | SE/DSE | LNE | SNNE | WSNNE | SAR |
> | --- | --- | --- | --- | --- | --- | --- | --- | --- | --- | --- |
> | **Llama-2-7b** | Multi-turn QA | CoQA | **0.7339** | 0.7316 | 0.6471 | 0.7284 | 0.6979 | 0.7208 | 0.7235 | 0.7148 |
> | Llama-2-7b | Math | MATH-500 | 0.6958 | 0.6910 | 0.5518 | **0.7026** | 0.6111 | 0.6775 | 0.6714 | 0.6924 |
> | Llama-2-7b | Coding | HumanEval | **0.6268** | 0.6209 | 0.4931 | 0.5564 | 0.5152 | 0.5983 | 0.5965 | 0.6010 |
> | Llama-2-7b | Summarization | CNN Dailymail | **0.6880** | 0.6794 | 0.4276 | 0.6539 | 0.5596 | 0.6697 | 0.6641 | 0.6324 |
> | Llama-2-7b | —————— | **Average** | **0.6861** | 0.6807 | 0.5299 | 0.6603 | 0.5960 | 0.6666 | 0.6639 | 0.6602 |
> |  |  |  |  |  |  |  |  |  |  |  |
> | **Llama-3-8b** | Multi-turn QA | CoQA | **0.7529** | 0.7441 | 0.5907 | 0.7433 | 0.6159 | 0.7472 | 0.7448 | 0.7276 |
> | Llama-3-8b | Math | MATH-500 | 0.7071 | 0.6994 | 0.6247 | **0.7104** | 0.6031 | 0.6951 | 0.6975 | 0.6974 |
> | Llama-3-8b | Coding | HumanEval | **0.6217** | 0.6173 | 0.5072 | 0.5822 | 0.4918 | 0.6117 | 0.6158 | 0.6103 |
> | Llama-3-8b | Summarization | CNN Dailymail | **0.6359** | 0.6200 | 0.4373 | 0.6040 | 0.5482 | 0.6133 | 0.6089 | 0.5904 |
> | Llama-3-8b | —————— | **Average** | **0.6794** | 0.6702 | 0.5400 | 0.6600 | 0.5648 | 0.6668 | 0.6668 | 0.6564 |
> |  |  |  |  |  |  |  |  |  |  |  |
> | **Qwen3-8B** | Multi-turn QA | CoQA | 0.7788 | **0.7797** | 0.6482 | 0.7715 | 0.6540 | 0.7613 | 0.7642 | 0.7509 |
> | Qwen3-8B | Math | MATH-500 | 0.7319 | 0.7298 | 0.4770 | **0.7375** | 0.6466 | 0.6900 | 0.6981 | 0.7225 |
> | Qwen3-8B | Coding | HumanEval | **0.6479** | 0.6352 | 0.4736 | 0.5734 | 0.5834 | 0.6387 | 0.6331 | 0.6381 |
> | Qwen3-8B | Summarization | CNN Dailymail | **0.6880** | 0.6794 | 0.4276 | 0.6539 | 0.5596 | 0.6697 | 0.6641 | 0.6453 |
> | Qwen3-8B | —————— | **Average** | **0.7117** | 0.7060 | 0.5066 | 0.6841 | 0.6109 | 0.6899 | 0.6899 | 0.6892 |

---

> > ### Author Response · Authors · 2025-11-21
> > **Response to Reviewer i66o (Part 2/2)**
> >
> > **Q3: Significance of the performance.** The performance of sampling-based uncertainty quantification depends on the stochastic nature of sampling. It would be great if the authors could report the variance/std of each approach, especially when their performance is close to Eigenscore.
> >
> > **A3**: Thanks for your kind suggestion. We acknowledge that adding variance/std in results can further justify the superiority of our method, and we reproduce all experiments in Table 1 **for 5 times** to calculate their mean and std values for each result. The results revised in Table 1 (copied below) further validate the robustness and superiority of our method over baselines under different random seeds.
> >
> > | Model | Dataset | ER (Ours) | ES | PF | DSE | LNE | SE |
> > | --- | --- | --- | --- | --- | --- | --- | --- |
> > | Llama-2-7b | TriviaQA | **0.7873±0.0022** | 0.7792±0.0019 | 0.6650±0.0025 | 0.7754±0.0015 | 0.6935±0.0015 | 0.7746±0.0012 |
> > | Llama-2-7b | SQuAD | **0.7211±0.0018** | 0.7197±0.0021 | 0.6613±0.0021 | 0.7176±0.0016 | 0.6519±0.0008 | 0.7179±0.0016 |
> > | Llama-2-7b | BioASQ | **0.8453±0.0016** | 0.8438±0.0011 | 0.7322±0.0023 | 0.8433±0.0019 | 0.4706±0.0031 | 0.8425±0.0012 |
> > | Llama-2-7b | NQ | 0.7041±0.0011 | **0.7064±0.0013** | 0.6589±0.0022 | 0.6966±0.0013 | 0.6500±0.0012 | 0.6996±0.0026 |
> > | Llama-2-7b | **Average** | **0.7645** | 0.7623 | 0.6794 | 0.7582 | 0.6165 | 0.7587 |
> > |  |  |  |  |  |  |  |  |
> > | Llama-2-13b | TriviaQA | **0.7412±0.0025** | 0.7344±0.0019 | 0.7356±0.0033 | 0.7328±0.0021 | 0.6927±0.0020 | 0.7371±0.0028 |
> > | Llama-2-13b | SQuAD | 0.7271±0.0024 | 0.7246±0.0027 | 0.6935±0.0027 | **0.7366±0.0020** | 0.6894±0.0020 | 0.7364±0.0014 |
> > | Llama-2-13b | BioASQ | **0.8234±0.0020** | 0.8213±0.0021 | 0.6951±0.0023 | 0.7943±0.0012 | 0.6894±0.0018 | 0.7992±0.0016 |
> > | Llama-2-13b | NQ | **0.7283±0.0015** | 0.7268±0.0014 | 0.6831±0.0027 | 0.7246±0.0016 | 0.6872±0.0021 | 0.7248±0.0026 |
> > | Llama-2-13b | **Average** | **0.7550** | 0.7517 | 0.7018 | 0.7471 | 0.6897 | 0.7494 |
> > |  |  |  |  |  |  |  |  |
> > | Mistral-7b | TriviaQA | **0.7635±0.0017** | 0.7575±0.0018 | 0.7200±0.0026 | 0.7579±0.0012 | 0.6521±0.0025 | 0.7559±0.0023 |
> > | Mistral-7b | SQuAD | 0.7214±0.0021 | 0.7133±0.0017 | 0.6381±0.0038 | 0.7222±0.0019 | 0.6223±0.0017 | **0.7236±0.0021** |
> > | Mistral-7b | BioASQ | **0.8562±0.0024** | 0.8525±0.0015 | 0.7178±0.0024 | 0.8505±0.0013 | 0.5946±0.0017 | 0.8508±0.0026 |
> > | Mistral-7b | NQ | 0.7662±0.0025 | 0.7624±0.0015 | 0.7531±0.0027 | **0.7675±0.0020** | 0.6779±0.0014 | 0.7658±0.0020 |
> > | Mistral-7b | **Average** | **0.7768** | 0.7714 | 0.7073 | 0.7745 | 0.6367 | 0.7740 |
> >
> > ---
> >
> > **Q4**: The current experiments were mainly conducted on short-form QA. I wonder whether ER can be applied to long-form QA, which is a more realistic setting.
> >
> > **A4**: Thank you for raising this point. First, we would like to clarify that the benchmarks we used are the common practice in the literature of hallucination detection, and these tasks remain unresolved for LLMs we evaluated. Thus, we respectfully consider that detecting hallucinations on these small-sized or open-source LLMs is still meaningful.
> >
> > However, we agree that detecting hallucination on more complex tasks is meaningful for evaluating our method. Following your suggestions, we additionally consider 4 more advanced datasets, including:
> >
> > - **CoQA (Multi-turn QA)**
> > - **MATH-500 (complex mathematical reasoning)**
> > - **HumanEval (code generation)**
> > - **CNN Dailymail (long-form summarization)**
> >
> > The results shown below validate the effectiveness of our method across these advanced datasets, showing its generalizability on complex reasoning tasks. We have added these results in Table 3 (shown above).
> >
> > ---
> >
> > ---
> >
> > We truly appreciate your valuable and detailed feedback. If you have any further questions or concerns, please let us know.

---

> ### Comment · Reviewer_i66o · 2025-11-21
> **No further concerns**
>
> I thank the authors for their clear and detailed responses and their revision. These addressed all my concerns, and I don't have further questions. I think I've already given a fair score, and I will keep my score accordingly.

---

> > ### Author Response · Authors · 2025-11-21
> > **Thank you for your positive feedback!**
> >
> > Dear Reviewer i66o,
> >
> > Thank you so much for your quick response and for keeping your positive review! Your helpful feedback is valuable to our research, and we're grateful for your contribution to the community.
> >
> > Best regards,
> >
> > Authors

---

### Official Review · Reviewer_uh3Q · 2025-10-28

**Soundness:** 2
**Presentation:** 2
**Contribution:** 2
**Rating:** 4
**Confidence:** 4

**Summary:**

This paper introduces a novel hallucination detection method for large language models (LLMs) based on the effective rank of hidden state embeddings. By analyzing the dispersion of embeddings across multiple layers and responses, the method quantifies uncertainty without requiring external tools or fine-tuning. Extensive experiments show that it performs competitively or better than existing baselines across diverse datasets and model architectures.

**Strengths:**

1. The paper introduces a spectral perspective (effective rank) for uncertainty quantification, which is effective.
2. The paper provides a clear theoretical motivation linking aleatoric and epistemic uncertainty to semantic divergence in hidden states.

**Weaknesses:**

1. Missing 2025 baseline, e.g., [1].
2. Llama-2 is old. It is recommended to conduct experiments on newer models and models from different series to verify the effectiveness and generalization of the proposed method fully.
3. While the proposed method is somewhat effective, measuring uncertainty through multiple sampling is not practical in real applications. Ten samplings means ten inferences, which is not very user-friendly for some long responses or even thinking-based LLMs.
4. The proposed method is essentially a measure of consistency and does not feel too novel.

Ref:
> [1] Beyond semantic entropy: Boosting LLM uncertainty quantification with pairwise semantic similarity

**Questions:**

See Weaknesses.

---

> ### Author Response · Authors · 2025-11-21
> **Response to Reviewer uh3Q (Part 1/2)**
>
> Dear Reviewer uh3Q,
>
> Thank you for your valuable feedback and recognizing our work as effective and clear. In response to your concerns, we have carefully revised the manuscript and provide detailed responses below.
>
> ---
>
> **Q1**: Missing 2025 baseline, e.g., [1].
>
> **A1**: Thank you for bringing [1] to our attention, which is an excellent recent work on hallucination detection. The difference between [1] and our method is that while [1] estimates the entropy-based uncertainty using nearest neighbor, our method quantifies uncertainty with effective rank, which are two distinct quantification paradigms. Following your suggestion, we additionally compared our method and both two versions of [1] (i.e. SNNE, WSNNE) in Table 3 (comparison under complex reasoning tasks), which demonstrates the superiority of our method. We copy the results from Table 3 below:
>
> | Model | Task | Dataset | ER (Ours) | ES | PF | SE/DSE | LNE | SNNE | WSNNE | SAR |
> | --- | --- | --- | --- | --- | --- | --- | --- | --- | --- | --- |
> | **Llama-2-7b** | Multi-turn QA | CoQA | **0.7339** | 0.7316 | 0.6471 | 0.7284 | 0.6979 | 0.7208 | 0.7235 | 0.7148 |
> | Llama-2-7b | Math | MATH-500 | 0.6958 | 0.6910 | 0.5518 | **0.7026** | 0.6111 | 0.6775 | 0.6714 | 0.6924 |
> | Llama-2-7b | Coding | HumanEval | **0.6268** | 0.6209 | 0.4931 | 0.5564 | 0.5152 | 0.5983 | 0.5965 | 0.6010 |
> | Llama-2-7b | Summarization | CNN Dailymail | **0.6880** | 0.6794 | 0.4276 | 0.6539 | 0.5596 | 0.6697 | 0.6641 | 0.6324 |
> | Llama-2-7b | —————— | **Average** | **0.6861** | 0.6807 | 0.5299 | 0.6603 | 0.5960 | 0.6666 | 0.6639 | 0.6602 |
> |  |  |  |  |  |  |  |  |  |  |  |
> | **Llama-3-8b** | Multi-turn QA | CoQA | **0.7529** | 0.7441 | 0.5907 | 0.7433 | 0.6159 | 0.7472 | 0.7448 | 0.7276 |
> | Llama-3-8b | Math | MATH-500 | 0.7071 | 0.6994 | 0.6247 | **0.7104** | 0.6031 | 0.6951 | 0.6975 | 0.6974 |
> | Llama-3-8b | Coding | HumanEval | **0.6217** | 0.6173 | 0.5072 | 0.5822 | 0.4918 | 0.6117 | 0.6158 | 0.6103 |
> | Llama-3-8b | Summarization | CNN Dailymail | **0.6359** | 0.6200 | 0.4373 | 0.6040 | 0.5482 | 0.6133 | 0.6089 | 0.5904 |
> | Llama-3-8b | —————— | **Average** | **0.6794** | 0.6702 | 0.5400 | 0.6600 | 0.5648 | 0.6668 | 0.6668 | 0.6564 |
> |  |  |  |  |  |  |  |  |  |  |  |
> | **Qwen3-8B** | Multi-turn QA | CoQA | 0.7788 | **0.7797** | 0.6482 | 0.7715 | 0.6540 | 0.7613 | 0.7642 | 0.7509 |
> | Qwen3-8B | Math | MATH-500 | 0.7319 | 0.7298 | 0.4770 | **0.7375** | 0.6466 | 0.6900 | 0.6981 | 0.7225 |
> | Qwen3-8B | Coding | HumanEval | **0.6479** | 0.6352 | 0.4736 | 0.5734 | 0.5834 | 0.6387 | 0.6331 | 0.6381 |
> | Qwen3-8B | Summarization | CNN Dailymail | **0.6880** | 0.6794 | 0.4276 | 0.6539 | 0.5596 | 0.6697 | 0.6641 | 0.6453 |
> | Qwen3-8B | —————— | **Average** | **0.7117** | 0.7060 | 0.5066 | 0.6841 | 0.6109 | 0.6899 | 0.6899 | 0.6892 |
>
> These results consistently validate our method outperforms both SNNE and WSNNE [1] across multiple datasets and models. Due to time limitations, we will add comparison to these two baselines on other tasks (Table 1) in our next version. Thanks again for the suggestion!

---

> > ### Author Response · Authors · 2025-11-21
> > **Response to Reviewer uh3Q (Part 2/2)**
> >
> > **Q2**: Llama-2 is old. It is recommended to conduct experiments on newer models and models from different series to verify the effectiveness and generalization of the proposed method fully.
> >
> > **A2**: Thank you for your kind advice. To reproduce and align with results from existing baselines, we mainly adopt Llama-2 and other models in our previous version. Following your suggestions, we additionally incorporated Qwen-3, Llama-3 for evaluation in Table 3 (as shown above), which consistently validates the effectiveness and generalization of our method across different models.
> >
> > ---
> >
> > **Q3**: While the proposed method is somewhat effective, measuring uncertainty through multiple sampling is not practical in real applications. Ten samplings means ten inferences, which is not very user-friendly for some long responses or even thinking-based LLMs.
> >
> > **A3**: Thanks for the insightful thought. We respectfully argue that while requiring additional computation, multiple sampling is actually practical under this setting, based on the following reasons:
> >
> > - Many state-of-the-art hallucination detection methods rely on multiple generation, especially representation-based methods, including most of baselines we compared (e.g. SNNE&WSNNE [1], Eigenscore, etc.). This requirement indicates a trade-off for hallucination detection effectiveness v.s. generation times.
> > - Current popular ranking-based decoding paradigms intrinsically needs multiple generation for preference ranking [R1, R2], offering unique opportunity for multiple generation based hallucination detection.
> >
> > [R1] Language Ranker: A Lightweight Ranking framework for LLM Decoding. NeurIPS 2025
> >
> > [R2] Advancing llm safe alignment with safety representation ranking. arXiv 2025
> >
> > ---
> >
> > **Q4**: The proposed method is essentially a measure of consistency and does not feel too novel.
> >
> > **A4**: Thank you for your insightful comment. We acknowledge that our method shares some conceptual similarities with hidden state-based uncertainty quantification approaches, but we respectfully argue that the effective rank introduces a more interpretable and unified framework for this line of research. Specifically, effective rank conveys human-readable metrics (quantified as the numerical number of clusters) while it stands fundamentally apart from prior work such as Eigenscore, which only delivers uninterpretable numerical outputs for classification purposes. Beyond its intuitive interpretability, our method also outperforms existing baselines in both efficiency and effectiveness, offering comprehensive improvements over the state-of-the-art. Thus, we respectfully contend that conceptual overlap does not fully warrant the critique of insufficient novelty.
> >
> > Ref:
> >
> > > [1] Beyond semantic entropy: Boosting LLM uncertainty quantification with pairwise semantic similarity
> > >
> >
> > ---
> >
> > We truly appreciate your valuable and detailed feedback. If you have any further questions or concerns, please let us know.

---

### Official Review · Reviewer_madp · 2025-11-01

**Soundness:** 2
**Presentation:** 3
**Contribution:** 3
**Rating:** 6
**Confidence:** 3

**Summary:**

This paper proposes a lightweight, training-free method for hallucination detection in large language models (LLMs) based on the effective rank of hidden-state embeddings. The core idea is that hallucinations correspond to higher semantic divergence in the model’s internal representations across multiple sampled responses and intermediate layers. By constructing an embedding matrix from these hidden states and computing its effective rank—defined as the exponential of the Shannon entropy of its singular values—the method yields an interpretable uncertainty score that reflects the “effective number of distinct semantic modes” in the model’s behavior. The authors provide both theoretical justification (via decomposition of aleatoric and epistemic uncertainty) and extensive empirical validation across four QA datasets and three LLMs (Llama-2-7B/13B, Mistral-7B). Results show that the proposed approach achieves competitive or superior AUROC compared to recent baselines—including Semantic Entropy and Eigenscore—while being computationally efficient and requiring no external tools or fine-tuning. Ablation studies further demonstrate robustness to choices of layer depth, number of samples, and temperature.

**Strengths:**

- **Clear Spectral Approach**: The paper introduces a lightweight, theoretically-motivated effective rank metric for detecting model uncertainty, rooted in spectral properties of internal representations. The method is both simple and interpretable.
- **Solid Empirical Validation Across Models and Tasks**: Experiments on three diverse LLMs (Llama-2-7b-chat, Llama-2-13b-chat, Mistral-7B) and multiple datasets (TriviaQA, NQ, BioASQ, SQuAD) show the method is robust and scalable (Table 1, Table 3).
- **Computational Efficiency**: Time benchmarks (Table 2) indicate negligible computational overhead, making the approach appealing for practical deployment.
- **Unification of Internal and External Uncertainty**: Theoretical analysis (Section 5) thoughtfully decomposes hidden state uncertainty into aleatoric and epistemic components, motivating the use of multiple sampled outputs and internal representations for hallucination detection.

**Weaknesses:**

###

- **Limited Novelty Relative to Prior Work**: The core idea of quantifying hallucination through dispersion or uncertainty in internal hidden states has clear precedents in recent literature—most notably in Eigenscore (Chen et al., 2024), which also leverages internal representations to measure model inconsistency. While the use of effective rank offers a novel spectral perspective, it shares substantial conceptual and methodological overlap with these prior approaches, as both operate on hidden-state covariance structures and aim to capture semantic uncertainty through representation diversity. Consequently, the advance appears incremental rather than fundamentally distinct.

- **Experiment Focuses on Factoid QA, Neglects Complex or Long-Form Generation**: All experiments are confined to **short-form, factoid QA tasks** (TriviaQA, NQ, BioASQ, SQuAD). The method’s applicability to more complex settings—such as multi-turn dialogue, long-form generation, or reasoning-intensive tasks—is untested. Given that hallucination manifests differently in these regimes (e.g., narrative drift, logical inconsistency), the generalizability of the approach as a “new paradigm” remains unsubstantiated.
- **Modest Performance Gains Over Strong Internal Baselines**: Despite winning on some datasets and models (see Table 1), the method is only modestly better or competitive with Eigenscore and SE/DSE; The improvement over the best prior internal-embedding metric is marginal (average AUROC gains are often under 0.005), raising questions about the practical significance of the advance. A more thorough analysis of when and why ER succeeds or fails relative to other baselines would strengthen the contribution claim.
- **Lack of In-Depth Failure Analysis**: While the paper includes ablation studies on factors like temperature (e.g., Table 4), it offers limited qualitative analysis of failure cases. In particular, there is insufficient discussion of two critical scenarios: (i) **low Effective Rank (ER) but hallucinated outputs**—which may arise from confidently held false knowledge—and (ii) **high ER but factually correct generations**—which could reflect legitimate ambiguity or paraphrasing. Such cases are essential for understanding the method’s blind spots and practical reliability. A deeper error analysis (e.g., categorizing failure modes or visualizing representation clusters) would significantly strengthen the paper’s empirical contribution and trustworthiness claims.

**Questions:**

- **Generalizability Beyond Factoid QA**: How does the effective rank approach perform for open-ended, long-form QA, or dialog tasks? Is it robust to hallucinations arising from multi-hop or contextual failures rather than surface-level entity mismatches?
- **Adaptive Layer Selection**: The ablation studies show that the optimal hidden layer for computing effective rank varies across models, datasets, and temperatures. Is there a practical, automatic strategy—e.g., based on prompt features, layer-wise consistency, or lightweight meta-learning—to select the best layer (or layer ensemble) without ground-truth labels?
- **Applicability to Black-Box/Closed Models**: The method requires access to internal embeddings; is there any potential for black-box adaptation, or is it confined to white-box settings?

---

> ### Author Response · Authors · 2025-11-21
> **Response to Reviewer n54Y (Part 1/3)**
>
> Dear Reviewer madp,
>
> Thank you for your valuable feedback and recognizing our work as clear, solid, and unified. In response to your concerns, we have carefully revised the manuscript and provide detailed responses below.
>
> ---
>
> **W1**: **Limited Novelty Relative to Prior Work**: The core idea of quantifying hallucination through dispersion or uncertainty in internal hidden states has clear precedents in recent literature—most notably in Eigenscore (Chen et al., 2024), which also leverages internal representations to measure model inconsistency. While the use of effective rank offers a novel spectral perspective, it shares substantial conceptual and methodological overlap with these prior approaches, as both operate on hidden-state covariance structures and aim to capture semantic uncertainty through representation diversity. Consequently, the advance appears incremental rather than fundamentally distinct.
>
> **A1**: Thank you for your insightful comment. We acknowledge that our method shares some conceptual similarities with hidden state-based uncertainty quantification approaches, but we respectfully argue that the effective rank introduces a more interpretable and unified framework for this line of research. Specifically, effective rank conveys human-readable metrics (quantified as the number of clusters) while it stands fundamentally apart from prior work such as Eigenscore, which only delivers uninterpretable numerical outputs for classification purposes. Beyond its intuitive interpretability, our method also outperforms existing baselines in both efficiency and effectiveness, offering comprehensive improvements over the state-of-the-art. Thus, we respectfully contend that conceptual overlap does not fully warrant the critique of insufficient novelty.
>
> ---
>
> **W2 & Q1: Experiment Focuses on Factoid QA, Neglects Complex or Long-Form Generation**: All experiments are confined to **short-form, factoid QA tasks** (TriviaQA, NQ, BioASQ, SQuAD). The method’s applicability to more complex settings—such as multi-turn dialogue, long-form generation, or reasoning-intensive tasks—is untested. Given that hallucination manifests differently in these regimes (e.g., narrative drift, logical inconsistency), the generalizability of the approach as a “new paradigm” remains unsubstantiated.
>
> Generalizability Beyond Factoid QA: How does the effective rank approach perform for open-ended, long-form QA, or dialog tasks? Is it robust to hallucinations arising from multi-hop or contextual failures rather than surface-level entity mismatches?
> **A2**: Thank you for your kind suggestion. First, we would like to clarify that the benchmarks we used are the common practice in the literature of hallucination detection, and these tasks remain unresolved for LLMs we evaluated (e.g., their average accuracy ~40%, far from “solved problems”). Though these accuracies may be higher for commercial LLMs like GPT-5, we respectfully consider that detecting hallucinations on these small-sized or open-source LLMs is still meaningful.
>
> However, we agree that detecting hallucination on more complex tasks is meaningful for evaluating our method. Following your suggestions, we additionally consider 4 more advanced datasets, including:
>
> - **CoQA (Multi-turn QA)**
> - **MATH-500 (complex mathematical reasoning)**
> - **HumanEval (code generation)**
> - **CNN Dailymail (long-form summarization)**
>
> The results shown below validate the effectiveness of our method across these advanced datasets, showing its generalizability on complex reasoning tasks. We have added these results in Table 3 (copied below).

---

> > ### Author Response · Authors · 2025-11-21
> > **Response to Reviewer n54Y (Part 2/3)**
> >
> > | Model | Task | Dataset | ER (Ours) | ES | PF | SE/DSE | LNE | SNNE | WSNNE | SAR |
> > | --- | --- | --- | --- | --- | --- | --- | --- | --- | --- | --- |
> > | **Llama-2-7b** | Multi-turn QA | CoQA | **0.7339** | 0.7316 | 0.6471 | 0.7284 | 0.6979 | 0.7208 | 0.7235 | 0.7148 |
> > | Llama-2-7b | Math | MATH-500 | 0.6958 | 0.6910 | 0.5518 | **0.7026** | 0.6111 | 0.6775 | 0.6714 | 0.6924 |
> > | Llama-2-7b | Coding | HumanEval | **0.6268** | 0.6209 | 0.4931 | 0.5564 | 0.5152 | 0.5983 | 0.5965 | 0.6010 |
> > | Llama-2-7b | Summarization | CNN Dailymail | **0.6880** | 0.6794 | 0.4276 | 0.6539 | 0.5596 | 0.6697 | 0.6641 | 0.6324 |
> > | Llama-2-7b | —————— | **Average** | **0.6861** | 0.6807 | 0.5299 | 0.6603 | 0.5960 | 0.6666 | 0.6639 | 0.6602 |
> > |  |  |  |  |  |  |  |  |  |  |  |
> > | **Llama-3-8b** | Multi-turn QA | CoQA | **0.7529** | 0.7441 | 0.5907 | 0.7433 | 0.6159 | 0.7472 | 0.7448 | 0.7276 |
> > | Llama-3-8b | Math | MATH-500 | 0.7071 | 0.6994 | 0.6247 | **0.7104** | 0.6031 | 0.6951 | 0.6975 | 0.6974 |
> > | Llama-3-8b | Coding | HumanEval | **0.6217** | 0.6173 | 0.5072 | 0.5822 | 0.4918 | 0.6117 | 0.6158 | 0.6103 |
> > | Llama-3-8b | Summarization | CNN Dailymail | **0.6359** | 0.6200 | 0.4373 | 0.6040 | 0.5482 | 0.6133 | 0.6089 | 0.5904 |
> > | Llama-3-8b | —————— | **Average** | **0.6794** | 0.6702 | 0.5400 | 0.6600 | 0.5648 | 0.6668 | 0.6668 | 0.6564 |
> > |  |  |  |  |  |  |  |  |  |  |  |
> > | **Qwen3-8B** | Multi-turn QA | CoQA | 0.7788 | **0.7797** | 0.6482 | 0.7715 | 0.6540 | 0.7613 | 0.7642 | 0.7509 |
> > | Qwen3-8B | Math | MATH-500 | 0.7319 | 0.7298 | 0.4770 | **0.7375** | 0.6466 | 0.6900 | 0.6981 | 0.7225 |
> > | Qwen3-8B | Coding | HumanEval | **0.6479** | 0.6352 | 0.4736 | 0.5734 | 0.5834 | 0.6387 | 0.6331 | 0.6381 |
> > | Qwen3-8B | Summarization | CNN Dailymail | **0.6880** | 0.6794 | 0.4276 | 0.6539 | 0.5596 | 0.6697 | 0.6641 | 0.6453 |
> > | Qwen3-8B | —————— | **Average** | **0.7117** | 0.7060 | 0.5066 | 0.6841 | 0.6109 | 0.6899 | 0.6899 | 0.6892 |

---

> ### Author Response · Authors · 2025-11-21
> **Response to Reviewer n54Y (Part 3/3)**
>
> **W3: Modest Performance Gains Over Strong Internal Baselines**: Despite winning on some datasets and models (see Table 1), the method is only modestly better or competitive with Eigenscore and SE/DSE; The improvement over the best prior internal-embedding metric is marginal (average AUROC gains are often under 0.005), raising questions about the practical significance of the advance. A more thorough analysis of when and why ER succeeds or fails relative to other baselines would strengthen the contribution claim.
>
> **A3**: Thank you for your kind suggestion. First, we respectfully note that the advancement of our method is not simply a gain in the AUROC performance; as we discussed in **A1** above, our method has intrinsic interpretability and notable improvement in efficiency. Additionally, in our newly incorporated advanced datasets, our method consistently outperforms the baselines, showing its scalability to more complex reasoning tasks.
>
> As for a more thorough analysis of when and why ER succeeds or fails, please refer to our response to **W4** below.
>
> ---
>
> **W4: Lack of In-Depth Failure Analysis**: While the paper includes ablation studies on factors like temperature (e.g., Table 4), it offers limited qualitative analysis of failure cases. In particular, there is insufficient discussion of two critical scenarios: (i) **low Effective Rank (ER) but hallucinated outputs**—which may arise from confidently held false knowledge—and (ii) **high ER but factually correct generations**—which could reflect legitimate ambiguity or paraphrasing. Such cases are essential for understanding the method’s blind spots and practical reliability. A deeper error analysis (e.g., categorizing failure modes or visualizing representation clusters) would significantly strengthen the paper’s empirical contribution and trustworthiness claims.
>
> **A4**: Thank you for your insightful suggestion. We have included detailed case studies in **Appendix F.8**, where we take an in-depth study on the failure modes of our method. Below are the concrete conclusions and key insights from our failure analysis:
>
> - **Low ER + Hallucination**: This stems from **systematically incorrect internal knowledge encoded with overconfidence** (e.g., the model consistently conflates "republican officers" with "corrupt officers" for a SQuAD query). ER fails here because it only measures aleatoric uncertainty (stochasticity) rather than epistemic uncertainty (knowledge gaps), as tight embedding clusters (low ER) merely reflect consistent wrong outputs .
> - **High ER + Correctness**: This is driven by **legitimate semantic diversity (e.g., paraphrasing) or spurious distractors** (e.g., core answer "French" accompanied by irrelevant "Quebec" for the Canada language query). High ER here is not a false positive but a meaningful signal of ambiguity, with core correct outputs forming dominant embedding clusters and distractors as minor subclusters .
>
> This analysis clarifies our method’s boundaries while providing actionable guidance to enhance real-world reliability.
>
> ---
>
> **Q2**: Adaptive Layer Selection: The ablation studies show that the optimal hidden layer for computing effective rank varies across models, datasets, and temperatures. Is there a practical, automatic strategy—e.g., based on prompt features, layer-wise consistency, or lightweight meta-learning—to select the best layer (or layer ensemble) without ground-truth labels?
> **A5**: Thank you for the valuable suggestion. To better understand the impact on the layer selection, we have compared and visualized the performance of our method across different layers in Figure 2 in Section 4.3, where the middle and final layers consistently demonstrate better performance, showing the robustness of our method without carefully selecting specific layers. Therefore, we propose that randomly selecting a layer after the middle one (i.e., $l$-th layer that $l\ge [\frac L 2]$, $L$ is the total number of layers) is sufficient to achieve a good detection performance, without carefully choosing a specific layer.
>
> ---
>
> **Q3**: Applicability to Black-Box/Closed Models: The method requires access to internal embeddings; is there any potential for black-box adaptation, or is it confined to white-box settings?
>
> **A6**: Thank you for your careful consideration. Alongside with other feature-based hallucination methods, our method is also a white-box one, which we have acknowledged in Section 1 (in contribution statements, page 2).
>
> ---
>
> We truly appreciate your valuable and detailed feedback. If you have any further questions or concerns, please let us know.

---

> > ### Comment · Reviewer_madp · 2025-11-26
> >
> > Thanks for your responses. Some of my concerns are addressed. I'll leave my scores positive.

---

> > > ### Author Response · Authors · 2025-11-26
> > > **Thank you for your positive feedback!**
> > >
> > > Dear Reviewer madp,
> > >
> > > Thank you very much for keeping your positive review! We appreciate your feedback and are happy to address your concerns. If you have further comments, we're glad to discuss them with you.
> > >
> > > Best regards,
> > >
> > > Authors

---

### Official Review · Reviewer_n54Y · 2025-11-02

**Soundness:** 3
**Presentation:** 3
**Contribution:** 3
**Rating:** 6
**Confidence:** 5

**Summary:**

This paper proposes "Effective Rank-based Uncertainty," a novel, training-free method for hallucination detection. The core idea is to quantify uncertainty by measuring the semantic divergence of hidden states across multiple sampled responses, using the "effective rank" of the hidden state matrix as a proxy.

**Strengths:**

The method is elegant, grounded in spectral analysis, and the intuition that high rank (i.e., high semantic divergence) correlates with high uncertainty (hallucination) is compelling.

The paper demonstrates competitive performance against other response-level uncertainty-based baselines like Semantic Entropy and Eigenscore.

**Weaknesses:**

The following weakness is not significant. So I give a positive overall score.



W1: The evaluation is confined to simplistic tasks that do not reflect modern LLM use cases. The paper's entire empirical validation rests on short-form, factual question-answering datasets (TriviaQA, NQ, BioASQ, SQUAD), these tasks are arguably "solved" problems for modern LLMs and do not represent the frontier of the current hallucination challenge.

The critical applications where hallucination detection is most needed today are in long-form generation (e.g., summarization, report writing), complex reasoning (e.g., multi-step math problems), and agentic workflows (e.g., software development, tool use). In these scenarios, hallucinations manifest as subtle logical fallacies, fabricated evidence, inconsistent reasoning steps, or non-existent API calls, not just incorrect facts. It is  unclear whether this method can effectively capture hallucination in these more complex errors.



W2: In real-world applications, LLM outputs are often hundreds or thousands of tokens long. A hallucination (e.g., a fabricated statistic in the first paragraph of a summary, or a subtle bug in a code block) could occur very early in the generation.

The design choice in this paper limits the method to a response-level label ("this entire 2000-token output is a hallucination: Yes/No"), which is of little practical use. Modern applications require real-time, token-level detection to pinpoint where the model deviates from a faithful trajectory. The proposed method is incapable of providing this necessary granularity.



W3: Some existing works develop classifiers (both supervised and unsupervised) that operate directly on the contextualized embeddings (hidden states) of each token to predict hallucinations [1]. Comparing against such methods is essential. Without this comparison, it is impossible to know if this complex, multi-sample spectral approach is more effective than a simple, single-pass classifier trained on the hidden states themselves. At least, you should mention these works in Section 2.

[1] Unsupervised Real-time Hallucination Detection based on the Internal States of Large Language Models

I am open to increasing my score if the authors address these weaknesses

**Questions:**

Please refer to the previous section.

---

> ### Author Response · Authors · 2025-11-21
> **Response to Reviewer n54Y (Part 1/2)**
>
> Dear Reviewer n54Y,
>
> Thank you for your valuable feedback and recognizing our work as elegant, grounded, and compelling. In response to your concerns, we have carefully revised the manuscript and provide detailed responses below.
>
> ---
>
> **W1**: The evaluation is confined to simplistic tasks that do not reflect modern LLM use cases. The paper's entire empirical validation rests on short-form, factual question-answering datasets (TriviaQA, NQ, BioASQ, SQUAD), these tasks are arguably "solved" problems for modern LLMs and do not represent the frontier of the current hallucination challenge.
>
> The critical applications where hallucination detection is most needed today are in long-form generation (e.g., summarization, report writing), complex reasoning (e.g., multi-step math problems), and agentic workflows (e.g., software development, tool use). In these scenarios, hallucinations manifest as subtle logical fallacies, fabricated evidence, inconsistent reasoning steps, or non-existent API calls, not just incorrect facts. It is unclear whether this method can effectively capture hallucination in these more complex errors.
>
> **A1**: Thank you for raising this point. First, we would like to clarify that the benchmarks we used are the common practice in the literature of hallucination detection, and these tasks remain unresolved for LLMs we evaluated (e.g., their average accuracy ~40%, not “solved problems”). Though these accuracies may be higher for commercial LLMs like GPT-5, we respectfully consider that detecting hallucinations on these small-sized or open-source LLMs is still meaningful.
>
> However, we agree that detecting hallucination on more complex tasks is meaningful for evaluating our method. Following your suggestions, we additionally consider 4 more advanced datasets, including:
>
> - **CoQA (Multi-turn QA)**
> - **MATH-500 (complex mathematical reasoning)**
> - **HumanEval (code generation)**
> - **CNN Dailymail (long-form summarization)**
>
> The results shown below validate the effectiveness of our method across these advanced datasets, showing its generalizability on complex reasoning tasks. We have added these results to Table 3.
>
> | Model | Task | Dataset | ER (Ours) | ES | PF | SE/DSE | LNE | SNNE | WSNNE | SAR |
> | --- | --- | --- | --- | --- | --- | --- | --- | --- | --- | --- |
> | **Llama-2-7b** | Multi-turn QA | CoQA | **0.7339** | 0.7316 | 0.6471 | 0.7284 | 0.6979 | 0.7208 | 0.7235 | 0.7148 |
> | Llama-2-7b | Math | MATH-500 | 0.6958 | 0.6910 | 0.5518 | **0.7026** | 0.6111 | 0.6775 | 0.6714 | 0.6924 |
> | Llama-2-7b | Coding | HumanEval | **0.6268** | 0.6209 | 0.4931 | 0.5564 | 0.5152 | 0.5983 | 0.5965 | 0.6010 |
> | Llama-2-7b | Summarization | CNN Dailymail | **0.6880** | 0.6794 | 0.4276 | 0.6539 | 0.5596 | 0.6697 | 0.6641 | 0.6324 |
> | Llama-2-7b | —————— | **Average** | **0.6861** | 0.6807 | 0.5299 | 0.6603 | 0.5960 | 0.6666 | 0.6639 | 0.6602 |
> | **Llama-3-8b** | Multi-turn QA | CoQA | **0.7529** | 0.7441 | 0.5907 | 0.7433 | 0.6159 | 0.7472 | 0.7448 | 0.7276 |
> | Llama-3-8b | Math | MATH-500 | 0.7071 | 0.6994 | 0.6247 | **0.7104** | 0.6031 | 0.6951 | 0.6975 | 0.6974 |
> | Llama-3-8b | Coding | HumanEval | **0.6217** | 0.6173 | 0.5072 | 0.5822 | 0.4918 | 0.6117 | 0.6158 | 0.6103 |
> | Llama-3-8b | Summarization | CNN Dailymail | **0.6359** | 0.6200 | 0.4373 | 0.6040 | 0.5482 | 0.6133 | 0.6089 | 0.5904 |
> | Llama-3-8b | —————— | **Average** | **0.6794** | 0.6702 | 0.5400 | 0.6600 | 0.5648 | 0.6668 | 0.6668 | 0.6564 |
> | **Qwen3-8B** | Multi-turn QA | CoQA | 0.7788 | **0.7797** | 0.6482 | 0.7715 | 0.6540 | 0.7613 | 0.7642 | 0.7509 |
> | Qwen3-8B | Math | MATH-500 | 0.7319 | 0.7298 | 0.4770 | **0.7375** | 0.6466 | 0.6900 | 0.6981 | 0.7225 |
> | Qwen3-8B | Coding | HumanEval | **0.6479** | 0.6352 | 0.4736 | 0.5734 | 0.5834 | 0.6387 | 0.6331 | 0.6381 |
> | Qwen3-8B | Summarization | CNN Dailymail | **0.6880** | 0.6794 | 0.4276 | 0.6539 | 0.5596 | 0.6697 | 0.6641 | 0.6453 |
> | Qwen3-8B | —————— | **Average** | **0.7117** | 0.7060 | 0.5066 | 0.6841 | 0.6109 | 0.6899 | 0.6899 | 0.6892 |

---

> > ### Author Response · Authors · 2025-11-21
> > **Response to Reviewer n54Y (Part 2/2)**
> >
> > **W2**: In real-world applications, LLM outputs are often hundreds or thousands of tokens long. A hallucination (e.g., a fabricated statistic in the first paragraph of a summary, or a subtle bug in a code block) could occur very early in the generation.
> >
> > The design choice in this paper limits the method to a response-level label ("this entire 2000-token output is a hallucination: Yes/No"), which is of little practical use. Modern applications require real-time, token-level detection to pinpoint where the model deviates from a faithful trajectory. The proposed method is incapable of providing this necessary granularity.
> >
> > **A2**: Thank you for raising this point. We agree that token-level detection of hallucination is a new paradigm for long content generation quality, but the current hallucination detection works are all in the response-level, which we consider still meaningful, as we discussed in A1 above. Specifically, we agree that LLM outputs can be very long in current days, but checking the faithfulness in short context tasks remains valuable and unresolved, and our method advances the state-of-the-art for this thread of work. Meanwhile, our additional evaluation on advanced datasets validates its versatility in long context hallucination detection. We agree that your consideration on *token-level detection* is a promising future research direction, but we respectively note that this may be out of the scope of this paper.
> >
> > ---
> >
> > **W3**: Some existing works develop classifiers (both supervised and unsupervised) that operate directly on the contextualized embeddings (hidden states) of each token to predict hallucinations [1]. Comparing against such methods is essential. Without this comparison, it is impossible to know if this complex, multi-sample spectral approach is more effective than a simple, single-pass classifier trained on the hidden states themselves. At least, you should mention these works in Section 2.
> >
> > **A3**: Thank you for bringing these related works to attention. We agree that [1] is an excellent work on hallucination detection with trained classifiers, but as pointed out in the introduction, these methods require external knowledge and additional training, as well as an additional explicit module during inference. By contrast, our method requires only the direct computation of the effective rank during inference, and we mainly compare it with baselines that do not use such external entities for fairness.
> >
> > Thus,  we have included discussion on classifier-based methods like [1] in related work (Section 2). Due to time limitations, we will include experimental comparisons with them in our camera-ready version. Thanks again for your suggestion!
> >
> > ---
> >
> > We truly appreciate your valuable and detailed feedback. If you have any further questions or concerns, please let us know.

---

### Author Response · Authors · 2025-11-29
**Final Remarks for Area Chair**

Dear Area Chair,

We sincerely appreciate your stepping in to oversee the review process for our submission under the unusual circumstances. We recognize the significant additional workload this reassignment imposes and are grateful for your time and effort in evaluating our work, the original reviews, and our revisions.

---

**Summary of Strengths**

Reviewers highlighted the following key strengths of our work:

- **Elegant & Theoretically Grounded:** The effective rank-based method provides a novel spectral perspective that is mathematically elegant and theoretically motivated, unifying internal and external uncertainty (Reviewers n54Y, uh3Q, madp).
- **Simple & Efficient:** The approach is training-free, computationally efficient (negligible overhead), and easy to deploy compared to classifier-based or NLI-based methods (Reviewers madp, i66o).
- **Effective Empirical Performance:** The method demonstrates robust performance across diverse models and datasets, often outperforming strong baselines (Reviewers n54Y, madp, i66o).
- **Clear Motivation:** The theoretical decomposition of aleatoric and epistemic uncertainty provides clear justification for using multiple sampled responses (Reviewers uh3Q, i66o).

---

**Response to Key Concerns**

**1. Generalization to Complex Tasks**

- **Concern:** The original evaluation was limited to short-form QA (TriviaQA, NQ, etc.), potentially failing to capture hallucinations in complex scenarios like long-form generation, reasoning, or coding (Reviewers n54Y, madp, i66o).
- **Resolution:** We significantly expanded our evaluation to include four advanced datasets: **CoQA** (Multi-turn QA), **MATH-500** (Complex Reasoning), **HumanEval** (Code Generation), and **CNN Dailymail** (Summarization). Our Effective Rank (ER) method consistently outperforms baselines on average on these tasks, demonstrating strong generalization to complex settings.
- **Location:** Section 4.2 / Table 3, Page 7.

**2. Baselines Comparison**

- **Concern:** Lacked comparisons to 2025 baselines (e.g., SNNE, WSNNE) and needed a clearer distinction from the conceptually similar method (Reviewers n54Y, uh3Q, madp, i66o).
- **Resolution:**
    - *Empirically*, we added experimental comparisons against **SNNE**, **WSNNE**, and **SAR** in Table 3, showing our method's consistent superiority over these very recent baselines.
    - *Conceptually*, although these baselines share certain similarities with the uncertainty–based methods, they have different focuses on uncertainty (pairwise similarity for SNNE/WSSNE  and uncertainty entropy corrected by attention for SAR). Regarding baselines that require external knowledge or modules (e.g., MIND), we added a discussion on them in the Related Work. Since these work need extra entities, we do not compare with purely uncertainty-based methods for fairness.
    - *Theoretically*, we added a detailed theoretical comparison in Appendix D.1, clarifying that while baselines like Eigenscore approximates differential entropy (often yielding negative, uninterpretable scores), ER provides a distinct, interpretable metric representing the "effective number of semantic categories".
- **Location:** Section 2.1, Page 3, Table 3, Page 7; Appendix D.1, Page 19.

**3. Evaluation on Modern Models**

- **Concern:** Experiments relied on older models; validation on newer LLMs was requested (Reviewers uh3Q, i66o).
- **Resolution:** We incorporated evaluations using **Llama-3-8b** and **Qwen3-8B**. The results confirm that our method's effectiveness holds and potentially scales better with these newer, more capable models.
- **Location:** Table 3, Page 7.

**4. Robustness and Failure Analysis**

- **Concern:** Reviewers requested analysis of failure modes and statistical significance (Reviewers madp, i66o).
- **Resolution:**
    - We repeated the main experiments (Table 1) five times and reported **Mean ± Std**, confirming the statistical robustness of our gains.
    - We added an in-depth **Failure Case Analysis** based on the sample analyses we conducted earlier in Appendix F.8, categorizing errors into "overconfident internal knowledge errors" (Low ER + Hallucination) and "legitimate semantic diversity" (High ER + Correctness), providing transparency on method limitations.
- **Location:** Table 1, Page 6; Appendix F.8, Page 27.

**5. Layer Selection Strategy**

- **Concern:** The optimal layer for analysis seemed to vary (Reviewer madp).
- **Resolution:** Building on our previous ablation experiments, which only selected the middle or final 1/5 of the hidden layers, we additionally analyzed the hallucination detection ability of each hidden layer and visualized it in **Figure 3**. Results indicate that middle-to-late layers generally yield stable and optimal performance across tasks. We updated our recommendation to prioritize these layers for practical deployment.
- **Location:** Appendix E.1 / Figure 3, Pages 20-21.

---

### Meta-Review · Area_Chair_AHVN · 2026-01-12

**Summary:**

This paper proposes a hallucination detection method based on measuring the effective rank of hidden representations across layers and multiple sampled outputs of an LLM. The approach leverages the spectral properties of internal states to quantify uncertainty without requiring additional models or external supervision, and is motivated by a theoretical analysis of internal and external uncertainty. The authors provide extensive empirical evaluations showing that the method generalizes across tasks and settings.

During the rebuttal, the authors addressed most of the reviewers’ concerns, but one major issue remains. The core idea of quantifying hallucinations through dispersion or uncertainty in internal hidden states has clear precedents in recent literature—most notably Eigenscore (Chen et al., 2024), which also leverages internal representations to measure model inconsistency. While the use of effective rank offers a novel spectral perspective, it shares substantial conceptual and methodological overlap with prior approaches, as both operate on hidden-state covariance structures and aim to capture semantic uncertainty through representation diversity. In addition, the method is only modestly better than, or competitive with, Eigenscore (Chen et al., 2024), which further raises doubts about the significance of the contribution. Consequently, the advance appears incremental rather than fundamentally distinct. Taken together, we recommend rejecting this work.

**Reviewer Concerns:**

Most of the concerns have been addressed while the major remains. The primary limitation of this work is that its core idea—detecting hallucinations via dispersion or uncertainty in internal hidden representations—substantially overlaps with existing approaches such as Eigenscore. Also, the marginal empirical improvement of this method over Eigenscore further raises doubts about the significance of the contribution.

**Reviewer Scores:**

Reviewer n54Y (score: 6) would potentially maintain the score.

Reviewer madp (score: 6) confirms the same score.

Reviewer i66o (score: 8) confirms the same score.

Reviewer uh3Q (score: 4) would potentially maintain the same score as the concerns are not well addressed.

---

### Decision · Program_Chairs · 2026-01-26

Reject